EMBO
Molecular Medicine

# LETM1 couples mitochondrial DNA metabolism and nutrient preference

Romina Durigon[1], Alice L Mitchell[1,†], Aleck WE Jones[1,†], Andreea Manole[2,3], Mara Mennuni[1], Elizabeth MA Hirst[4], Henry Houlden[2,3], Giuseppe Maragni[5], Serena Lattante[5], Paolo Niccolo' Doronzio[5], Ilaria Dalla Rosa[1], Marcella Zollino[5], Ian J Holt[1,6,7] & Antonella Spinazzola[1,2,*]

## Abstract

The diverse clinical phenotypes of Wolf–Hirschhorn syndrome (WHS) are the result of haploinsufficiency of several genes, one of which, *LETM1*, encodes a protein of the mitochondrial inner membrane of uncertain function. Here, we show that LETM1 is associated with mitochondrial ribosomes, is required for mitochondrial DNA distribution and expression, and regulates the activity of an ancillary metabolic enzyme, pyruvate dehydrogenase. LETM1 deficiency in WHS alters mitochondrial morphology and DNA organization, as does substituting ketone bodies for glucose in control cells. While this change in nutrient availability leads to the death of fibroblasts with normal amounts of LETM1, WHS-derived fibroblasts survive on ketone bodies, which can be attributed to their reduced dependence on glucose oxidation. Thus, remodeling of mitochondrial nucleoprotein complexes results from the inability of mitochondria to use specific substrates for energy production and is indicative of mitochondrial dysfunction. However, the dysfunction could be mitigated by a modified diet—for WHS, one high in lipids and low in carbohydrates.

**Keywords** LETM1; mitochondrial DNA; mitochondrial morphology; nutrient utilization; Wolf–Hirschhorn syndrome

**Subject Categories** Genetics, Gene Therapy & Genetic Disease; Metabolism

## Introduction

Mitochondria are major contributors to cellular energy production, converting nutrients to ATP through the oxidative phosphorylation system (OXPHOS). Thirteen essential proteins of OXPHOS are transcribed from mitochondrial DNA (mtDNA) and synthesized by the mitochondrial ribosomes (mitoribosomes), via a highly specialized system that is tightly coupled to the inner mitochondrial membrane (IMM; Liu & Spremulli, 2000). In yeasts, the proposed machinery of co-translational membrane insertion includes Oxa1, mba1, and mdm38 (Ott & Herrmann, 2010). Mdm38 is an integral inner membrane protein conserved among eukaryotes, and it is suggested to play a role in multiple facets of mitochondrial metabolism, the precise nature of which remains unclear. Under certain conditions, it co-fractionates with the mitoribosome, on buoyant density gradients (Frazier *et al*, 2006; Kehrein *et al*, 2015), and it is proposed to regulate mitochondrial translation (Bauerschmitt *et al*, 2010). Furthermore, mdm38 regulates mitochondrial morphology and volume by controlling ion homeostasis across the mitochondrial inner membrane (Nowikovsky *et al*, 2004, 2007; Froschauer *et al*, 2005).

The human mdm38 homolog, LETM1, is inferred to regulate mitochondrial volume in a similar way to its yeast counterpart, as its ablation results in swollen mitochondria (Dimmer *et al*, 2002, 2008; Hasegawa & van der Bliek, 2007; Tamai *et al*, 2008; Jiang *et al*, 2009; McQuibban *et al*, 2010), whereas elevated expression produces condensed mitochondria with compact cristae (Hasegawa & van der Bliek, 2007). Whether monoallelic *LETM1* deletion results in mitochondrial dysfunction is still uncertain, as some studies described an alteration of the assembly of the respiratory chain supercomplexes (Tamai *et al*, 2008), while others reported no evident mitochondrial dysfunction or link with the mitochondrial ribosomal machinery (Dimmer *et al*, 2008).

*LETM1* is often deleted in Wolf–Hirschhorn syndrome (WHS; Zollino *et al*, 2003), a disorder caused by deletions of the short arm of one chromosome 4, encompassing multiple genes. Although LETM1 levels are consistently decreased in fibroblasts and lymphoblastoid cell lines of WHS patients harboring *LETM1* deletions (Hasegawa & van der Bliek, 2007; Dimmer *et al*, 2008; Doonan

1   Department of Clinical and Movement Neurosciences, UCL Institute of Neurology, London, UK
2   MRC Centre for Neuromuscular Diseases, UCL Institute of Neurology and National Hospital for Neurology and Neurosurgery, London, UK
3   Department of Molecular Neuroscience, UCL Institute of Neurology, London, UK
4   MRC Mill Hill Laboratory, London, UK
5   Institute of Genomic Medicine, Catholic University, Rome, Italy
6   Biodonostia Health Research Institute, San Sebastián, Spain
7   IKERBASQUE, Basque Foundation for Science, Bilbao, Spain
    *Corresponding author. Tel: +44 20 77940500 ext 33009; Fax: +44 20 7472 682; E-mail: a.spinazzola@ucl.ac.uk
    †These authors contributed equally to this work

*et al*, 2014; Hart *et al*, 2014), only modest alterations of mitochondrial membrane potential and free radical production have been reported (Doonan *et al*, 2014; Hart *et al*, 2014). Moreover, mice with a single copy of *Letm1* appear normal, with none of the facial or midline developmental defects typical of WHS *LETM1*-deficient patients. However, pyruvate dehydrogenase (PDH) activity was decreased in the brain of fasted $Letm1^{+/-}$ mice, suggesting a role for the protein in glucose metabolism (Jiang *et al*, 2013). The uncertainty surrounding LETM1 has left a gap in our understanding of the role of LETM1 protein both in mitochondrial biology and in its clinical relevance to WHS. Therefore, we set out to address several of the outstanding questions: of the role of LETM1 in mitochondrial gene expression in mammals; whether mitochondrial dysfunction in WHS is attributable to *LETM1* haploinsufficiency; and whether the effect on glucose metabolism in humans is comparable to the $Letm1^{+/-}$ mouse (Jiang *et al*, 2013). Analysis of the form and function of mitochondria in WHS patient-derived fibroblasts and *LETM1* gene-silenced cells indicates that LETM1 couples pyruvate oxidation to mtDNA metabolism, and that *LETM1* deficiency in WHS results in mitochondrial dysfunction that exacerbates the disorder.

# Results

## LETM1 is required for mitochondrial translation and respiration in mammalian cells

Findings from yeasts suggest that the LETM1 homolog, mdm38, facilitates the translation and assembly of specific mitochondrial proteins into respiratory chain complexes (Bauerschmitt *et al*, 2010). To address whether LETM1 contributed to mitochondrial protein synthesis (MPS) in mammals, we repressed its expression in HeLa cells via RNA interference. All three siRNAs targeting *LETM1* [siR1, siR2, or siR3 (Appendix Fig S1A)] caused mitochondrial swelling (Fig 1A), as previously reported (Dimmer *et al*, 2002, 2008; Hasegawa & van der Bliek, 2007; Tamai *et al*, 2008; Jiang *et al*, 2009; McQuibban *et al*, 2010), and each siRNA impaired MPS (Fig 1B and C; and Appendix Fig S1B) albeit to different extents. The marked decrease in LETM1 protein induced by siR1 or siR3

produced a more severe impairment of mitochondrial translation than siR2, which was associated with more residual protein (Fig 1B and C). The steady-state level of selected OXPHOS subunits was also directly proportional to the amount of LETM1 (Fig 1D), and the largest decreases in respiratory capacity were associated with the most pronounced reduction in LETM1 expression (Fig 1E). Thus, LETM1 deficiency impairs mitochondrial translation and restricts respiratory capacity in mammalian cells in a dose-dependent manner.

## LETM1 depletion compromises 55S ribosome maintenance and alters the levels and distribution of mtDNA and mtRNA

To determine the effect of LETM1 depletion on the mitochondrial ribosome, the abundance of selected mitochondrial ribosomal polypeptides and ribosomal RNAs was determined after *LETM1* knockdown. *LETM1* silencing decreased the levels of structural components of both mitoribosome subunits, MRPS17 and MRPL11, and the assembly factor C7orf30 (Rorbach *et al*, 2012; Wanschers *et al*, 2012; Fig 2A and Appendix Fig S1C), as well as the 12S and 16S ribosomal RNAs (Fig 2B). Furthermore, when lysates were fractionated on sucrose gradients the 55S ribosome was significantly decreased in cells partially depleted of LETM1, with a concomitant increase in the proportion of mitoribosome components at the top of the gradient (Fig 2C). These data suggest that LETM1 is required for 55S ribosome assembly or stability.

Mitochondrial transcription supplies the messenger and ribosomal RNAs for protein synthesis, and ribosomal (small subunit) assembly has been proposed to occur at the mitochondrial nucleoid (He *et al*, 2012a; Bogenhagen *et al*, 2014; Dalla Rosa *et al*, 2014). Moreover, RNA processing and ribosome assembly involve a suite of factors organized in RNA granules in close proximity to the mtDNA, including GRSF1 (Antonicka *et al*, 2013; Jourdain *et al*, 2013; Tu & Barrientos, 2015). Hence, mitoribosome dysfunction could be the cause or the consequence of aberrant mtRNA and mtDNA metabolism. Therefore, we next determined the effects of LETM1 depletion on the mitochondrial RNA and DNA levels. LETM1 deficiency consistently increased mtDNA copy number (Fig 2D), whereas it reduced the mitochondrial transcript

**Figure 1.  LETM1 is required for mitochondrial translation and respiration.**

A  Immunofluorescence analysis of HeLa cells transfected with either a non-target dsRNA (NT) or one of the three siRNAs targeting *LETM1* (siR1, siR2, or siR3) and labeled with anti-TOM20 antibody. In siR1-treated cells, the mitochondria formed a "honeycomb" of swollen distinct organelles; siR3 resulted in giant organelles with a central region distinguished by reduced TOM20; siR2 produced relatively little swelling, and the mitochondrial network was generally well preserved. The pronounced swelling induced by siR1 significantly increased circularity ($P$ = 2.32E-58; ImageJ analysis), siR1 circularity = 0.683 ± 0.012 compared with 0.414 ± 0.008 for the NT, where 1 = a perfect circle. Over 150 mitochondria were quantified in each cell type; data are mean ± SEM. Scale bar represents 12 μm in the main images and 4 μm in offset magnification.

B  $^{35}$S labeling of *de novo* mitochondrial protein synthesis of HeLa cells transfected as in (A). Polypeptide assignments flank the gel images. Coomassie-stained gels are used as loading controls, and immunoblots indicate the efficiency of *LETM1* knockdown.

C  Quantification of the radiolabeled mitochondrial polypeptides in panel (B) and similar gels, expressed relative to protein synthesis of the NT. Data are expressed as mean ± SEM of $n$ = 6 independent experiments; 1 or 2 rounds of transfection for siR1 and siR3, but exclusively 2 rounds in the case of siR2.

D  Representative immunoblots for the OXPHOS proteins NDUFB8 and COII of HeLa cells transfected with NT or si*LETM1* (siR1, siR2, or siR3). Vinculin and GAPDH are shown as loading controls. The mean relative abundances for respiratory subunits COII and NDUFB8 are shown beneath the blots. Data are expressed as mean ± SEM of $n$ = 8 independent experiments.

E  Mitochondrial oxygen consumption rate (OCR) measured using a Seahorse flux analyzer before (basal) and after the addition (maximal) of the uncoupler FCCP, in HeLa cells treated with NT or si*LETM1* (siR1 or siR2). Data are expressed as mean ± SEM of $n$ = 5 independent experiments.

Data information: Unpaired *t*-test (panel A) and one-way ANOVA in panels (B–E). *P*-values < 0.05 were considered to be statistically significant and labeled as follows: *$P$ < 0.05, **$P$ < 0.01, and ***$P$ < 0.001. ns, not statistically significant.
Source data are available online for this figure.

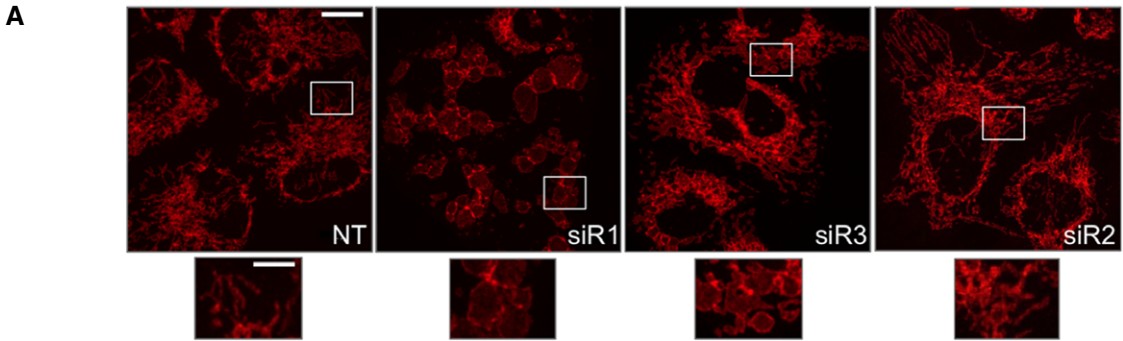

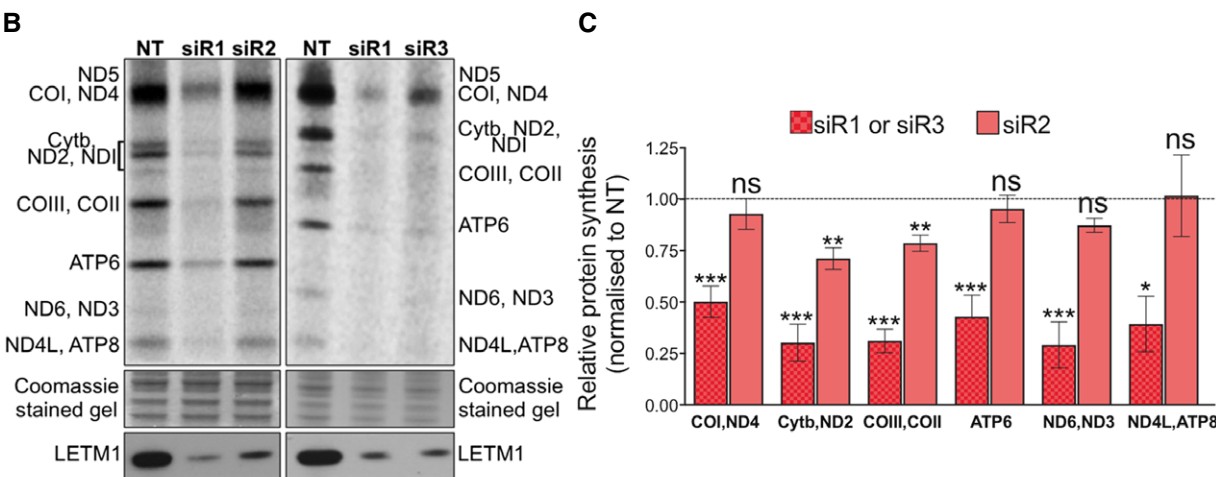

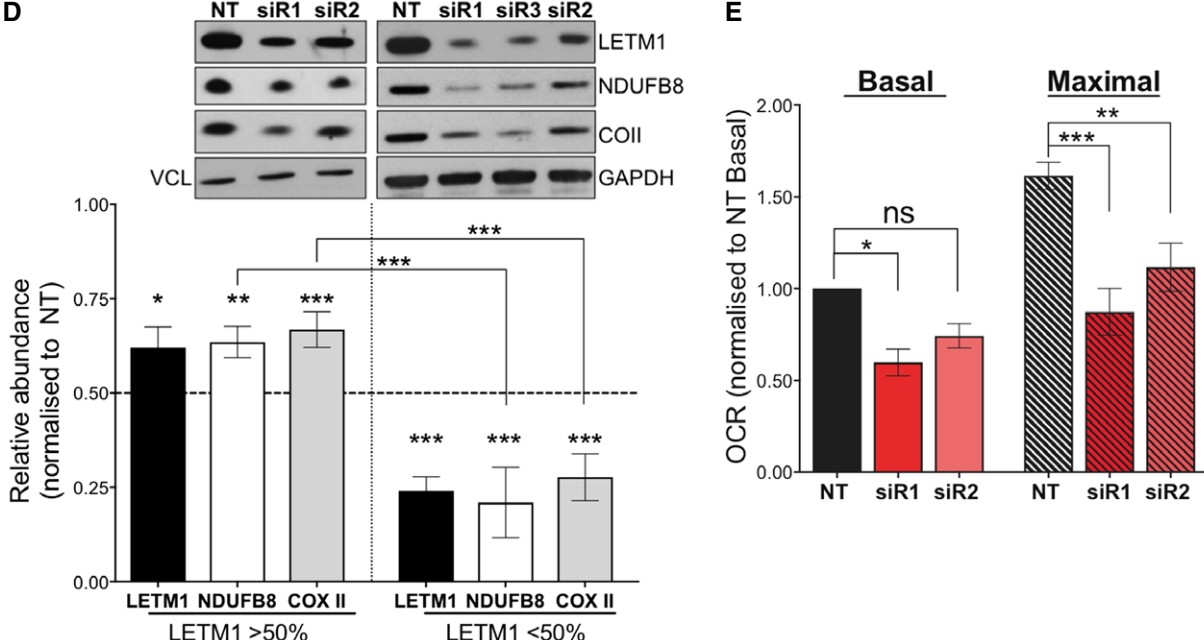

Figure 1.

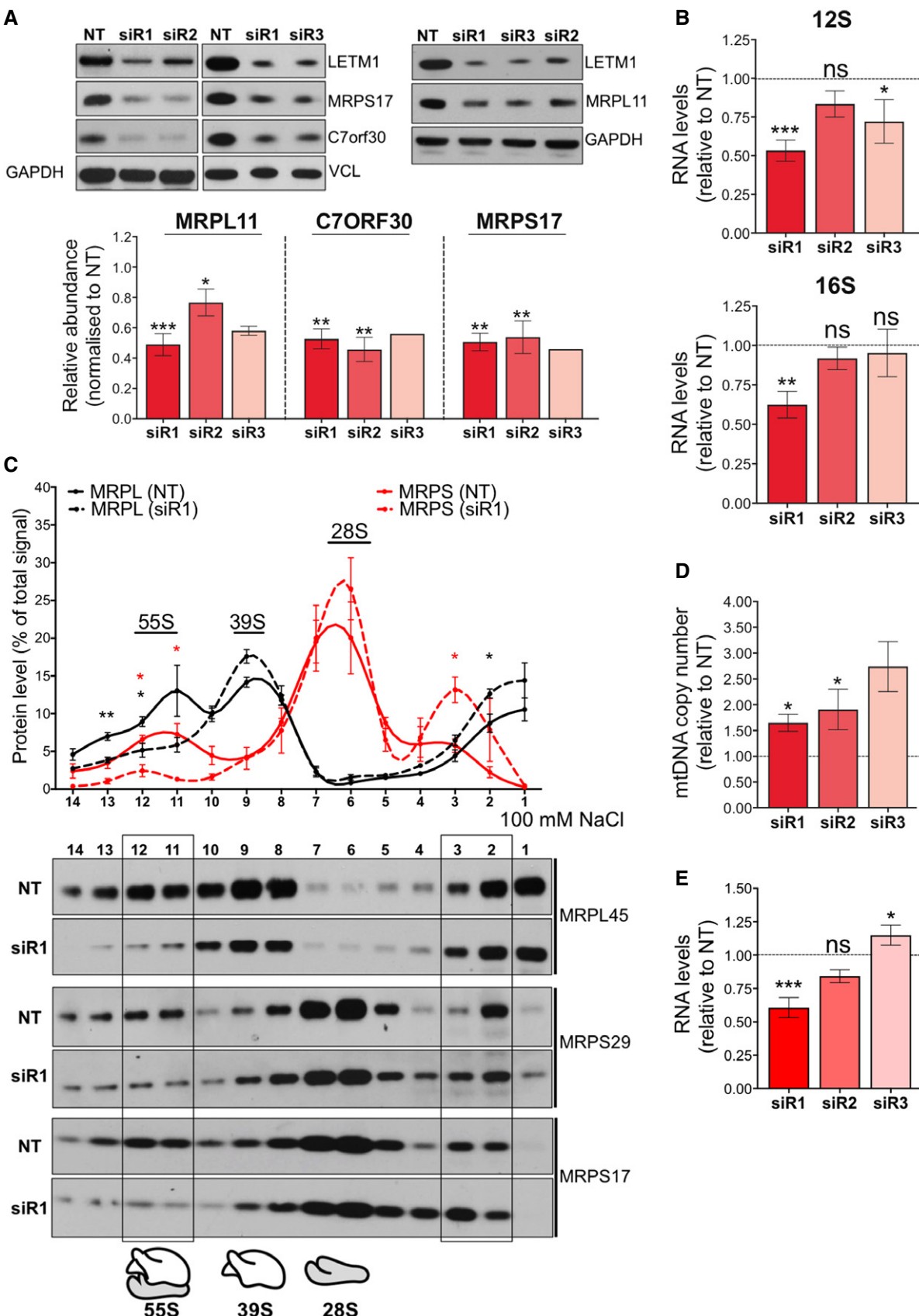

**Figure 2.**

◄

**Figure 2.  *LETM1* depletion compromises mitochondrial ribosome maintenance and alters the abundance of mitochondrial DNA and RNAs.**

A   Steady-state levels of mitochondrial ribosomal structural subunits (MRPL11 and MRPS17) or assembly factor (C7orf30) of HeLa cells transfected with siRNAs for either NT or *LETM1* (siR1, siR2, or siR3). Vinculin and GAPDH are shown as loading controls. Data are expressed as mean ± SEM of *n* = 3 independent experiments for siR1 and siR2, and *n* = 2 (MRPL11) or 1 (MRPS17 and C7orf30) for siR3.
B   Relative 12S (upper panel) and 16S rRNA (lower panel) levels in HeLa cells after *LETM1* siRNA compared to NT, *n* = 3 independent experiments.
C   Total lysates from HeLa cells treated with NT or a siRNA for *LETM1* were separated on 100 mM NaCl, 10–30% isokinetic sucrose gradients, and fractions analyzed by immunoblotting with antibodies to components of the large (39S) and small (28S) subunit of the 55S ribosome. Immunoblots were quantified by ImageJ, and the value for each fraction was expressed as a percentage of the sum of all fractions. Data are expressed as mean ± SEM of *n* = 3 independent experiments.
D   Relative mtDNA copy number estimated by qPCR. Data are expressed as mean ± SEM of *n* = 3 independent experiments except for siR3 (*n* = 2).
E   The combined average mRNA levels of ND1, COII, COIII, Cyt b, and ATP6/8 in HeLa cells treated with either NT or si*LETM1* (siR1, siR2, or siR3). Data are mean values ± SEM of *n* = 3 independent experiments.

Data information: One-way ANOVA in panels (A, B, D, and E); Unpaired *t*-test with Welch's correction in panel (C). *P*-values < 0.05 are considered to be statistically significant and labeled as follows: \**P* < 0.05, \*\**P* < 0.01, and \*\*\**P* < 0.001. ns, not statistically significant.
Source data are available online for this figure.

abundance per mtDNA (Fig 2E and Appendix Fig S1D). The highest mtDNA copy number, 2.6 times that of controls with siR3, was accompanied by transcript levels similar to controls, while the more modest 1.5-fold increase in mtDNAs induced by siR1 was associated with significantly reduced mRNA levels (Fig 2D and E). Thus, LETM1 deficiency impairs mitochondrial transcription, compensated, fully in the case of siR3, by elevated mtDNA copy number.

In light of the perturbations to mitochondrial DNA and RNA levels (Fig 2D and E), we next determined the effects of *LETM1* silencing on mtDNA and RNA form and distribution. All three siRNAs targeting *LETM1* produced mtDNA alterations (Figs 3A and B, and EV1A; and Appendix Fig S2), characterized by an increase in the size of many mtDNA foci (in the case of siR1; Fig 3A-ii) or by clustering of mtDNA (siR3 and siR2), the extent of which, again, correlated with the level of LETM1 protein (Figs 3A-iii, 4A-iv, and EV1A; and Appendix Fig S2). Similarly, the foci formed by newly synthesized RNA and an RNA granule protein, GRSF1, were aberrant in size and distribution in *LETM1*-deficient cells (Fig 3C and D).

**A fraction of LETM1 associates with mitochondrial nucleoprotein complexes**

To determine whether the phenotypes associated with LETM1 depletion reflected a direct interaction with the nucleic acids or ribosomes, we first assessed the pattern of LETM1 distribution in mitochondria. Immunofluorescence analysis of endogenous LETM1 indicated that the protein did not form the punctate patterns of either mitochondrial nucleoids (Spelbrink, 2010; Fig 4A) or newly synthesized RNA or GRSF1 (Antonicka *et al*, 2013; Jourdain *et al*, 2013; Tu & Barrientos, 2015; Fig EV1B), and was instead distributed throughout the mitochondrial network, comparable to mitochondrial ribosomes, and the outer mitochondrial membrane

protein TOM20 (Figs 4B and EV1B). Hence, immunofluorescence lacked the resolution to determine whether LETM1 physically interacts with the mitoribosome; therefore, we next fractionated mitochondrial lysates on two types of density gradient. Mitochondrial ribosomes and nucleoids cluster in one or two fractions on iodixanol gradients, well separated from the majority of mitochondrial proteins (He *et al*, 2012b), and a portion of LETM1 co-fractionated with these nucleoprotein complexes (Fig 4C). LETM1 also co-affinity purified with mitochondrial DNA interacting proteins in a recent study (Matic *et al*, 2018). For the specific analysis of mitochondrial ribosomes via sucrose gradient sedimentation, different salt concentrations were tested, since it has been proposed that the interaction of the yeast mdm38 with ribosomes is a function of the ionic strength (Kehrein *et al*, 2015). The 55S mitochondrial ribosome of cultured cells readily dissociates into its constituent 28S and 39S subunits on sucrose gradients; nevertheless, the small amount of intact 55S ribosomes co-migrated with a fraction of the LETM1 protein when mitochondrial lysates were separated using a 50 mM, but not 100 mM NaCl buffer (Fig 4D). These findings suggest that LETM1 physically interacts with the human mitochondrial ribosome, which can include interaction with mRNA (activators), as has been proposed for the yeast homolog (Bauerschmitt *et al*, 2010). The effects of LETM1 deficiency on mitochondrial DNA and RNA as well as the 55S ribosome fit with the protein facilitating ribosome assembly, which occurs at the nucleoid (He *et al*, 2012a; Bogenhagen *et al*, 2014; Dalla Rosa *et al*, 2014) and closely associated RNA granules (Antonicka *et al*, 2013; Jourdain *et al*, 2013; Tu & Barrientos, 2015). On the other hand, yeast mitochondrial ribosomes and nucleoids are interconnected (Kehrein *et al*, 2015) and co-purification analysis is consistent with a similar arrangement in human mitochondria (Rorbach *et al*, 2008; He *et al*, 2012b). Thus, LETM1 deficiency could primarily disturb the connections between

**Figure 3.  *LETM1* repression perturbs mtDNA and mtRNA organization.**

A   *LETM1* expression was suppressed in HeLa cells by transfection with targeted si*RNAs* (siR1, siR2, or siR3). A non-target dsRNA (NT) served as control. Cells were fixed and immunolabeled with anti-DNA antibody (green). A higher magnification of selected mtDNA foci is shown beside each picture.
B   Quantification of cells in (A) displaying mtDNA abnormalities. At least 50 cells per siRNA were counted from 4 (siR2) and 5 (siR1 or siR3) independent experiments. Data are expressed as mean ± SEM. \*\*\**P* < 0.001 (one-way ANOVA).
C   HeLa cells were labeled with bromouridine (BrU) for 60 mins, 144 h after transfection with NT or siR2, and stained with anti-BrdU (green) and anti-GRSF1 (red) antibodies. Cell nuclei were stained with DAPI (blue).
D   HeLa cells were labeled with anti-DNA (green) and anti-GRSF1 (red) antibodies, 144 h after transfection with NT or siR3. Cell nuclei were stained blue with DAPI.

Data information: Scale bars represent 12 μm in the main images and 4 μm in offset magnifications.
Source data are available online for this figure.

▶

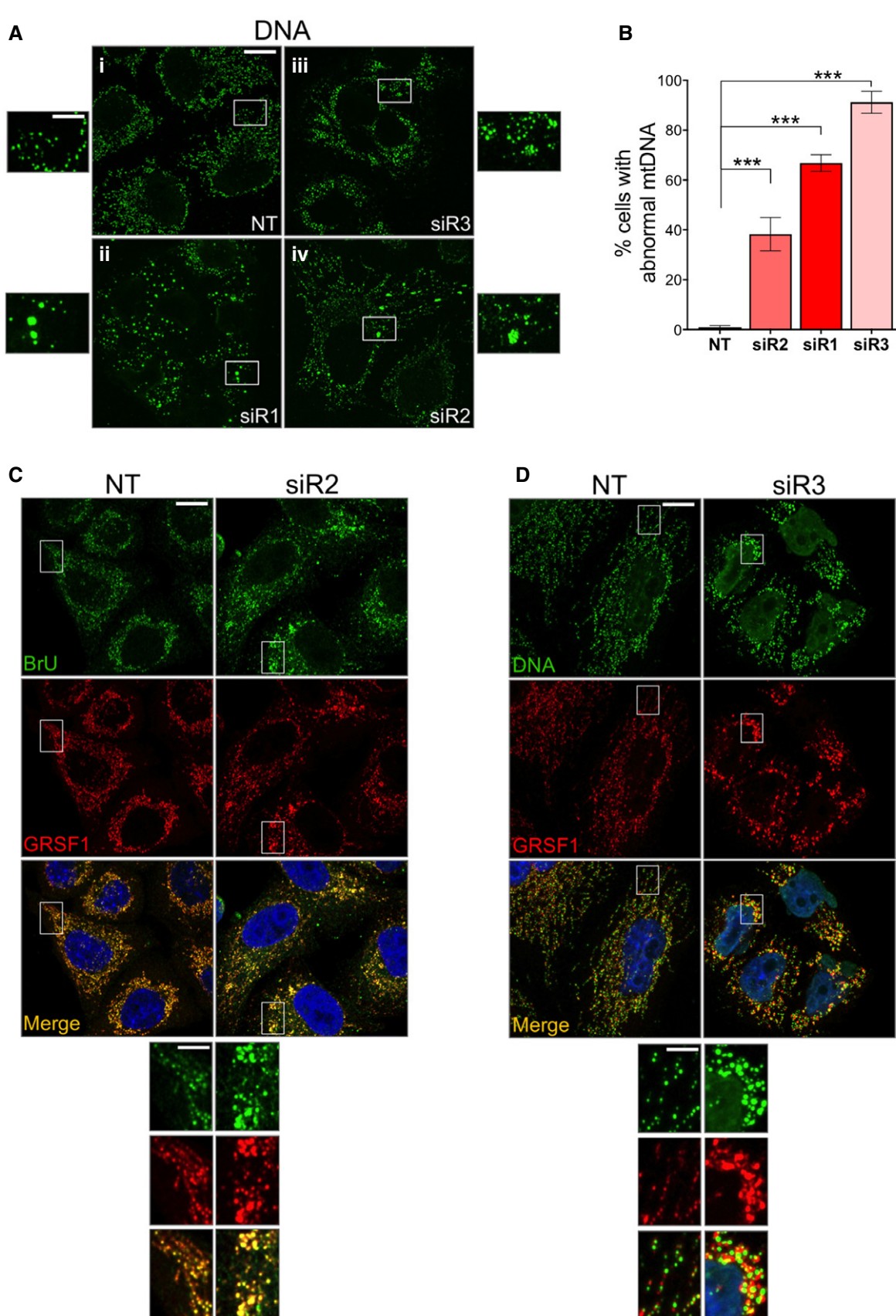

**Figure 3.**

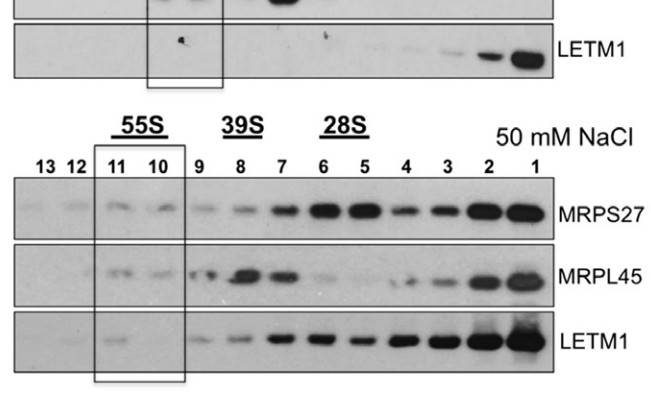

**Figure 4.**

**Figure 4.  LETM1 co-fractionates with mitochondrial nucleoprotein complexes and is widely distributed in the mitochondrial network, like ribosomes and unlike mtDNA.**

A, B    HeLa cells were immunostained with anti-DNA (green) and anti-LETM1 (red) (A), or anti-LETM1 (green) and anti-MRPS18 (red) (B).

C    Mitochondrial lysates from HEK293T cells were fractionated on 20–42.5% iodixanol gradients and the migration patterns of LETM1 and selected nucleoid components determined by immunoblotting.

D    HeLa mitochondrial lysates were fractionated on 10–30% sucrose gradients, and the distribution of LETM1 and ribosomal proteins was determined by immunoblotting. 55S corresponds to the assembled mitoribosome and 28S and 39S to its small and large subunits, respectively. Mitochondria were lysed in 100 or 50 mM NaCl (top and bottom panels, respectively).

Data information: Scale bars are 12 μm in the main images and 3 μm in offset magnifications.

Source data are available online for this figure.

mitochondrial nucleoids and ribosomes resulting in impaired ribosome maintenance and protein synthesis, when severe. Whatever the details of the interdependence of mtDNA organization and expression, the findings for LETM1 suggest that insufficiency or defects in many components of mitochondrial nucleoids or the mitochondrial translation machinery could produce similar phenotypes; indeed, MPV17L2 depletion causes marked mtDNA aggregation and severe ribosome maintenance defects (Dalla Rosa *et al*, 2014), and a mutation in the aminoacyl tRNA synthetase, FARS2, causes mtDNA aggregation (Almalki *et al*, 2014).

### Decreased DRP1 activity mitigates the effects of reduced expression of LETM1

We next examined the mitochondrial fission factor DRP1 in the LETM1-deficient cells, because its repression is known to cause mtDNA clustering (Ban-Ishihara *et al*, 2013). Transfection of HeLa cells with siR1 increased the abundance of the active phosphorylated form of DRP1 (DRP1$^{S616}$; Fig 5A and Appendix Fig S3A), whereas the two LETM1 siRNAs (siR2 and siR3) associated with mtDNA clustering had decreased levels of (active) DRP1 protein (Fig 5A). Therefore, the mtDNA clustering associated with *LETM1* silencing might be dependent on DRP1 repression. Moreover, since siR1 produced the most severe impairment of mitochondrial translation, repressing DRP1 when LETM1 is scarce might improve mtDNA expression. Concordant with the latter idea, co-silencing of *DRP1* and *LETM1* attenuated the effects of *LETM1* silencing alone (siR1), both in respect of MPS and the steady-state level of selected respiratory chain components (Fig 5B and C; and Appendix Fig S3B). As well as enhancing the expression of OXPHOS complexes, the dual gene silencing altered the mtDNA, producing a phenotype

intermediate between that of either *DRP1* or *LETM1* repression alone (Fig 5D). Hence, silencing of *LETM1* and *DRP1* has different, possibly opposing, effects on mtDNA organization, which could explain the benefit of co-repressing DRP1 when there is a shortage of LETM1, as occurred spontaneously with siR3 (and on occasion siR2) targeting *LETM1* (Figs 3 and EV1A; and Appendix Fig S2) and in the context of WHS (see below).

### Loss of one allele of *LETM1* is associated with mitochondrial swelling and mtDNA clustering

The defined mitochondrial phenotypes caused by LETM1 deficiency in HeLa cells provided points of reference to assess the consequences of *LETM1* haploinsufficiency in WHS. Mitochondrial metabolism was investigated in primary fibroblasts of four subjects with a monoallelic deletion on the short arm of chromosome 4 encompassing *LETM1* (S1–S4, *LETM1*$^{+/-}$), each of whom presented with typical WHS features and epilepsy (Appendix Table S1 and Appendix Fig S4). Fibroblasts of a fifth subject, S5 (the mother of S3), whose deletion did not include *LETM1* (thus *LETM1*$^{+/+}$), were analyzed in parallel. Cells of S1–S4 expressed approximately half as much LETM1 protein as control cells or those of S5 (Figs 6A and EV2A). Hence, *LETM1* haploinsufficiency is associated with reduced expression of LETM1 protein in WHS fibroblasts, whereas the adjacent genes deleted in subject S5 had no impact on the expression of LETM1 (Appendix Fig S4). However, with this amount of residual LETM1 protein, mitochondrial abnormalities were expected to be at the milder end of the spectrum observed in HeLa cells silenced for *LETM1*. Accordingly, there was no appreciable impairment of mitochondrial translation (Appendix Fig S5A), and there was little or no decrease in respiratory chain components in WHS fibroblasts

**Figure 5.  Changes to DRP1 in response to LETM1 depletion.$^{§}$**

A    Representative immunoblots of LETM1, DRP1 total protein levels, and activating phosphorylation (DRP1$^{S616}$) in HeLa cells treated with either NT or *LETM1* siR1, siR2, or siR3. GADPH is shown as a loading control. Immunoblot signals were quantified using ImageJ; data are displayed as mean ± SEM of *n* = 4 independent experiments except for siR3 DRP1$^{S616}$ (*n* = 2).

B    One hour $^{35}$S pulse-labeling of mitochondrial translated proteins in HeLa cells silenced for *LETM1* (siR1) or *DRP1* (siD), or co-silenced for *LETM1* and *DRP1* (siR1D) for 144 h. Coomassie-stained gels are used as loading controls, and polypeptide assignments are indicated to the right.

C    Immunoblotting of cell lysates prepared as in (B) with antibodies to respiratory chain subunits NDUFB8 and COII, and GAPDH as loading control. Data are expressed as mean ± SEM of *n* = 3 independent experiments.

D    Confocal analysis of HeLa cells silenced for *LETM1* (siR1), or for *DRP1* (siD), or co-silenced with both siR1 and siD for 96 h. Mitochondrial network, mtDNA, and cell nuclei were immunostained with anti-TOM20 (red) anti-DNA (green), and DAPI (blue), respectively. Scale bars are 6 μm in the main images and 3 μm in offset merged magnifications.

Data information: One-way ANOVA in panels (A and C). *$P$ < 0.05, **$P$ < 0.01, ***$P$ < 0.001. ns, not statistically significant.

Source data are available online for this figure.

$^{§}$Correction added online on 2 August 2018 after first online publication: the labels "siD" and "siR1+siD" in Figure 5D have been corrected.

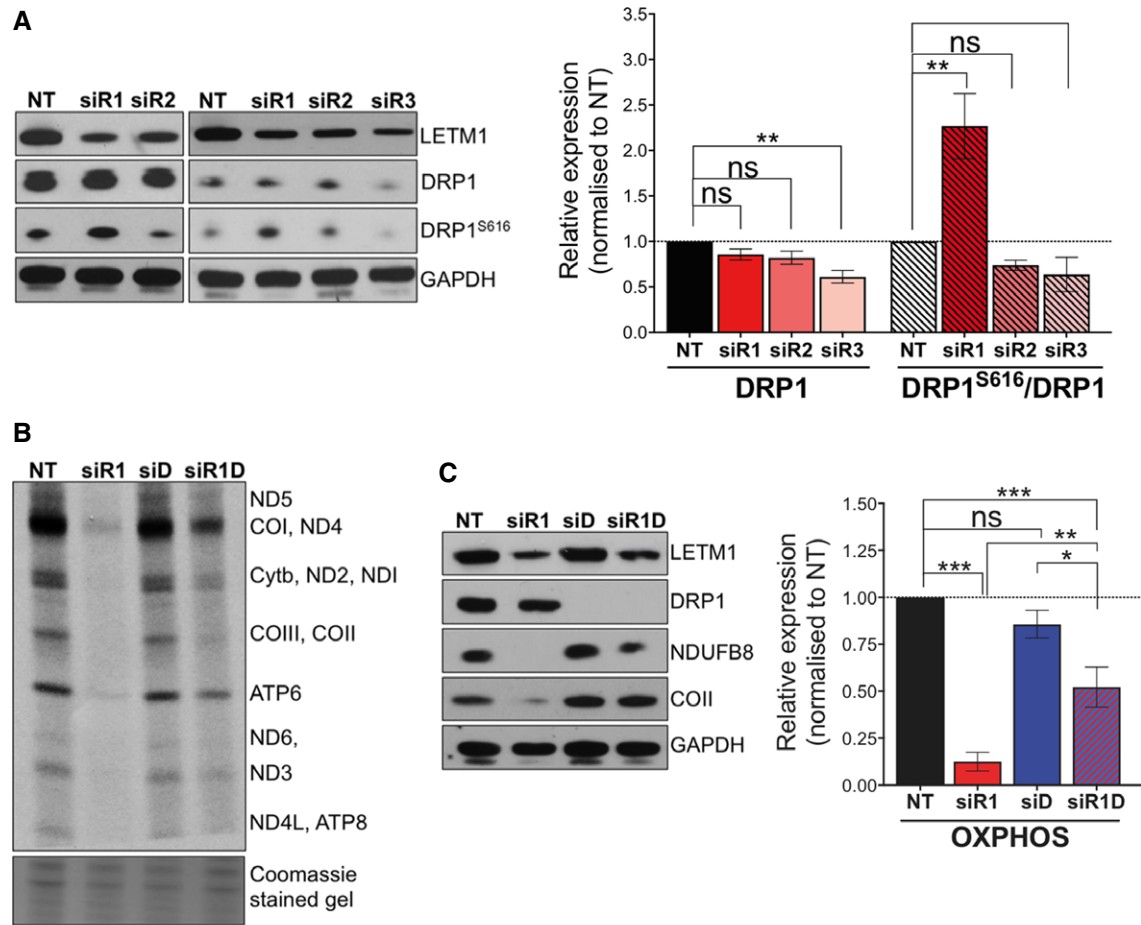

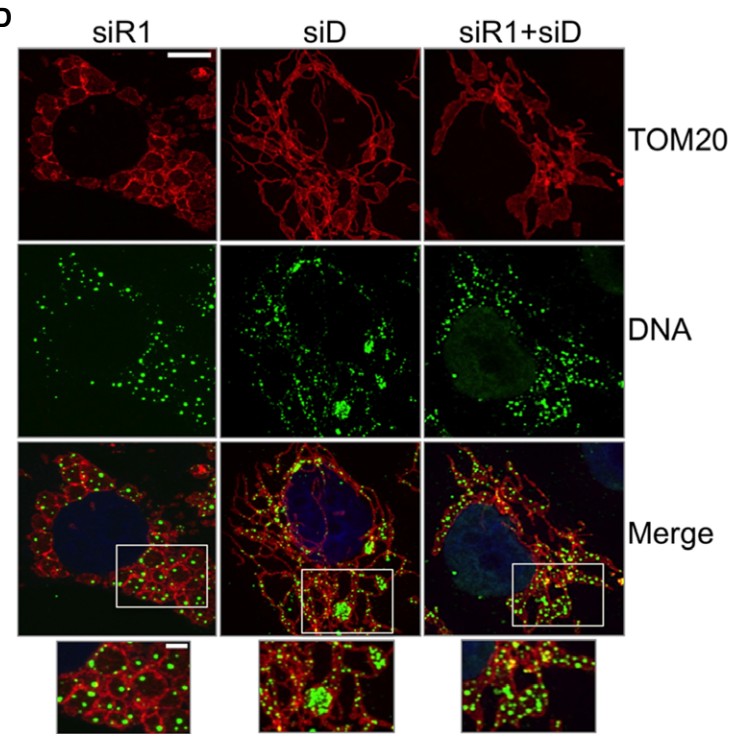

**Figure 5.**

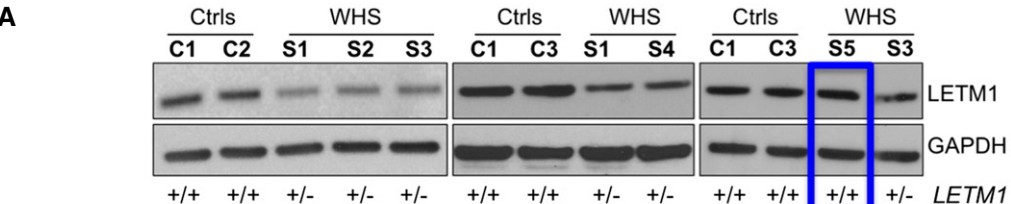

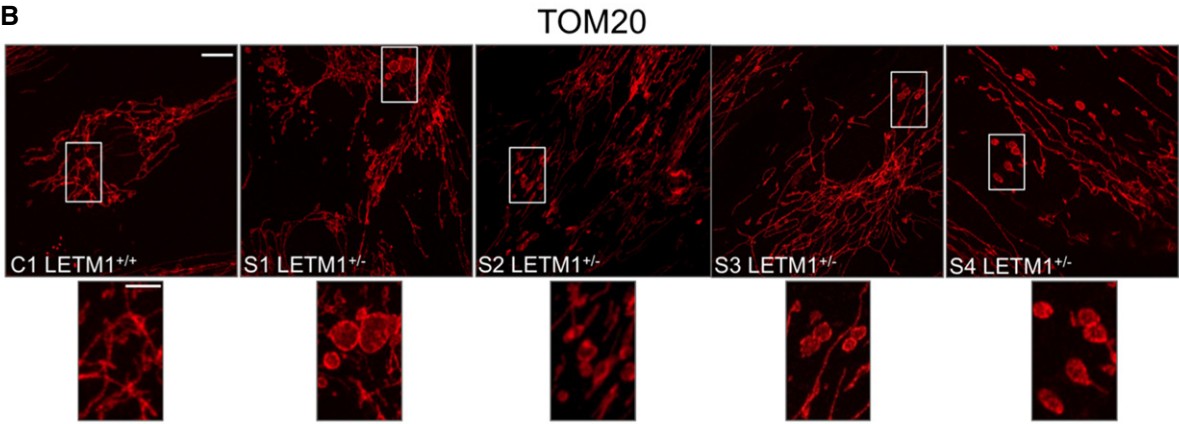

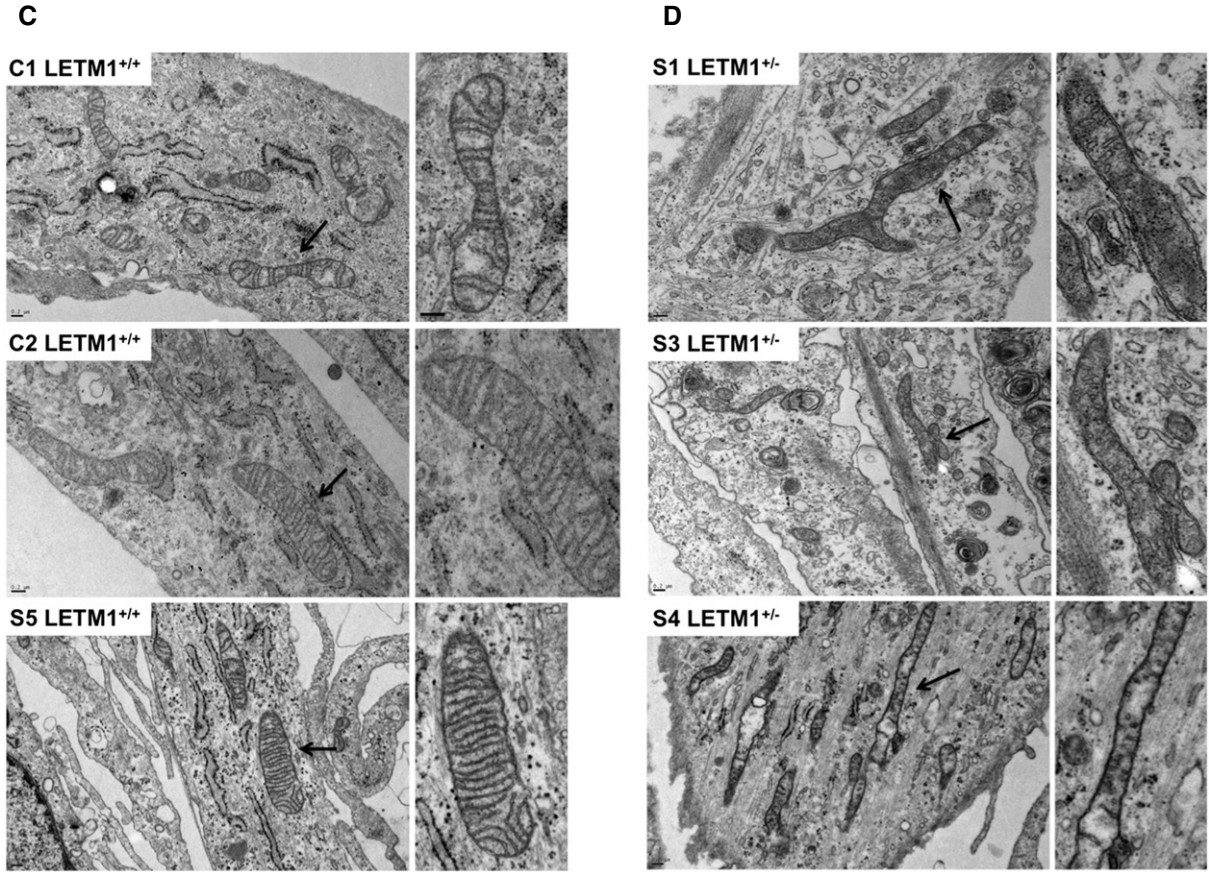

**Figure 6.**

**Figure 6.   LETM1 haploinsufficiency in WHS is associated with abnormal mitochondrial morphology and ultrastructure.**

A   Steady-state levels of LETM1 protein in fibroblasts from controls (C1–C3), WHS *LETM1*[+/−] (S1, S3, S4), and S5, the sole case of WHS in this study whose deletion does not encompass *LETM1* (boxed in blue). GAPDH is shown as a loading control.

B   Immunofluorescence analysis of control (C1), WHS *LETM1*[+/−] S1-S4, and fibroblasts with anti-TOM20 antibody. Scale bars are 8 μm in the main images and 3 μm in offset merged magnifications.

C   Electron micrographs of controls (C1 and C2) and WHS *LETM1*[+/+] (S5) cells; side panels show zooms. Scale bar: 0.2 μm. Black arrows indicate the areas shown offset at higher magnification.

D   Electron micrographs of WHS LETM1[+/−] S1, S3, and S4 cells. Black arrows indicate elongated and branched mitochondria; side panels show magnifications. Scale bar: 0.2 μm.

Source data are available online for this figure.

(Appendix Fig S5B); nevertheless, there were mitochondrial perturbations. Confocal microscopy revealed abnormal mitochondrial morphology in all the WHS cell lines lacking one copy of *LETM1*, and many mitochondria were more elongated than those of control cells and S5 (Appendix Fig S6A) and had numerous distensions (Fig 6B), while the largest mitochondria were often separated from the main network (Fig 6B zoom), suggesting that the swellings caused the mitochondrial network to fragment in places.

The abnormal mitochondrial morphology reported here was marked and not reported in previous studies of WHS fibroblasts (Hasegawa & van der Bliek, 2007; Dimmer *et al*, 2008; Doonan *et al*, 2014); therefore, we further analyzed mitochondrial ultrastructure in WHS fibroblasts by transmission electron microscopy. Fibroblasts from three controls (C1–C3) and subject S5 contained mitochondria of normal appearance (Figs 6C and EV2C). In contrast, approximately 70% of mitochondria of the *LETM1*[+/−] WHS patients were branched, thinned, and elongated (Fig 6D); frequently, the cristae were scarce and the remainder were rounded in shape (Figs 6D and EV2B–D), while the mitochondrial matrix in organelles where cristae were present appeared dense and more compact than normal (Fig EV2C). Hence, the loss of one allele of *LETM1* in fibroblasts of WHS subjects is associated with mitochondrial morphology abnormalities based on light and electron microscopy.

The distended and swollen mitochondria of the LETM1-deficient WHS fibroblasts displayed mtDNA clustering (Fig 7A and B; and Appendix Fig S6B) in a significantly greater proportion of cells than controls or subject S5 (Fig 7C), and this was associated with elevated mtDNA copy number in the cell lines tested (S1 and S4; Fig 7D). DRP1 was decreased in all cells of LETM1-deficient WHS subjects compared to controls, especially the active form of the protein (Fig 7E), which can account for the elongated mitochondria (Appendix Fig S6A). This finding strengthens the earlier conclusion that the repression of DRP1 is a frequent consequence of LETM1 deficiency that causes mtDNA clustering, and the data from HeLa cells suggest that it is a compensatory mechanism that limits the adverse effects of a shortage of LETM1 (Fig 5). Although the repression of DRP1 was associated with elongated mitochondria, there were also many discrete swollen mitochondria in the WHS cells (Fig 6B) suggesting that LETM1 deficiency causes partial fragmentation of the mitochondrial network in a DRP1-independent manner, as previously proposed (Dimmer *et al*, 2008). Taken together, these features indicate that WHS fibroblasts lacking one *LETM1* allele recapitulate features of synthetic *LETM1* deficiency in HeLa cells, including aberrant mitochondrial morphology, mtDNA disorganization, and elevated copy number.

### WHS patient fibroblasts remodel their metabolism away from pyruvate oxidation toward ketone body utilization

Mitochondrial morphology and dynamics are altered in response to a number of stimuli, including nutrient deprivation, and it has been proposed that changes in mitochondrial dynamics remodel substrate utilization for bioenergetic purposes (Buck *et al*, 2016; Wai & Langer, 2016). Hence, the mitochondrial morphology of the WHS fibroblasts lacking one *LETM1* allele might be indicative of a nutrient-restricted state, or increased reliance on substrates other than glucose. Moreover, the brain of *Letm1* heterozygous mice, under fasted conditions, displays a reduction in pyruvate dehydrogenase (PDH) activity (Jiang *et al*, 2013), which is critical for pyruvate utilization by mitochondria. Thus, we judged the regulation of PDH to be a prime potential function of LETM1. Analysis of iodixanol gradients of HEK cells indicated that almost all of the PDH-E1 subunit co-fractionated with mitochondrial nucleoprotein complexes (Fig 8A), consistent with its partner subunit (E2) being highly enriched in preparations of amphibian mtDNA (Bogenhagen *et al*, 2003), and its co-purification with mtDNA binding proteins and many components of the mitochondrial translation apparatus of HeLa cells (Matic *et al*, 2018). The inactive phosphorylated form of PDH-E1 was more abundant in WHS fibroblasts than in controls when cultured in 25 mM glucose and 1 mM pyruvate (Fig 8B and

**Figure 7.   WHS LETM1[+/−] fibroblasts display increased mtDNA copy number and mtDNA clusters.**

A   Confocal analysis of control C2, WHS *LETM1*[+/−] S4, and WHS *LETM1*[+/+] S5 cells, immunolabeled with anti-TOM20 (red), DNA (green), and DAPI (blue). Scale bar: 8 μm.

B   Confocal images of control C1 and WHS *LETM1*[+/−] S3 labeled with TOM20, DNA, and DAPI, shown as merge. Scale bar: 8 μm.

C   Quantification of cells in (A) displaying mtDNA aggregates. At least 50 cells per cell line were counted from three independent experiments. Data are expressed as mean ± SEM.

D   qPCR quantified mtDNA copy number in WHS and control cells. Values are expressed relative to control cells (C1–C3). Data are expressed as mean ± SEM of *n* = 3 independent experiments.

E   Relative abundance of DRP1 and DRP1[S616] in whole-cell extracts of control and WHS fibroblasts, with GAPDH shown as a loading control. Immunoblot signals were quantified using ImageJ, and data are expressed as mean ± SEM of *n* = 5 independent experiments.

Data information: One-way ANOVA in panels (C–E). **$P < 0.01$, ***$P < 0.001$. ns, not statistically significant.
Source data are available online for this figure.

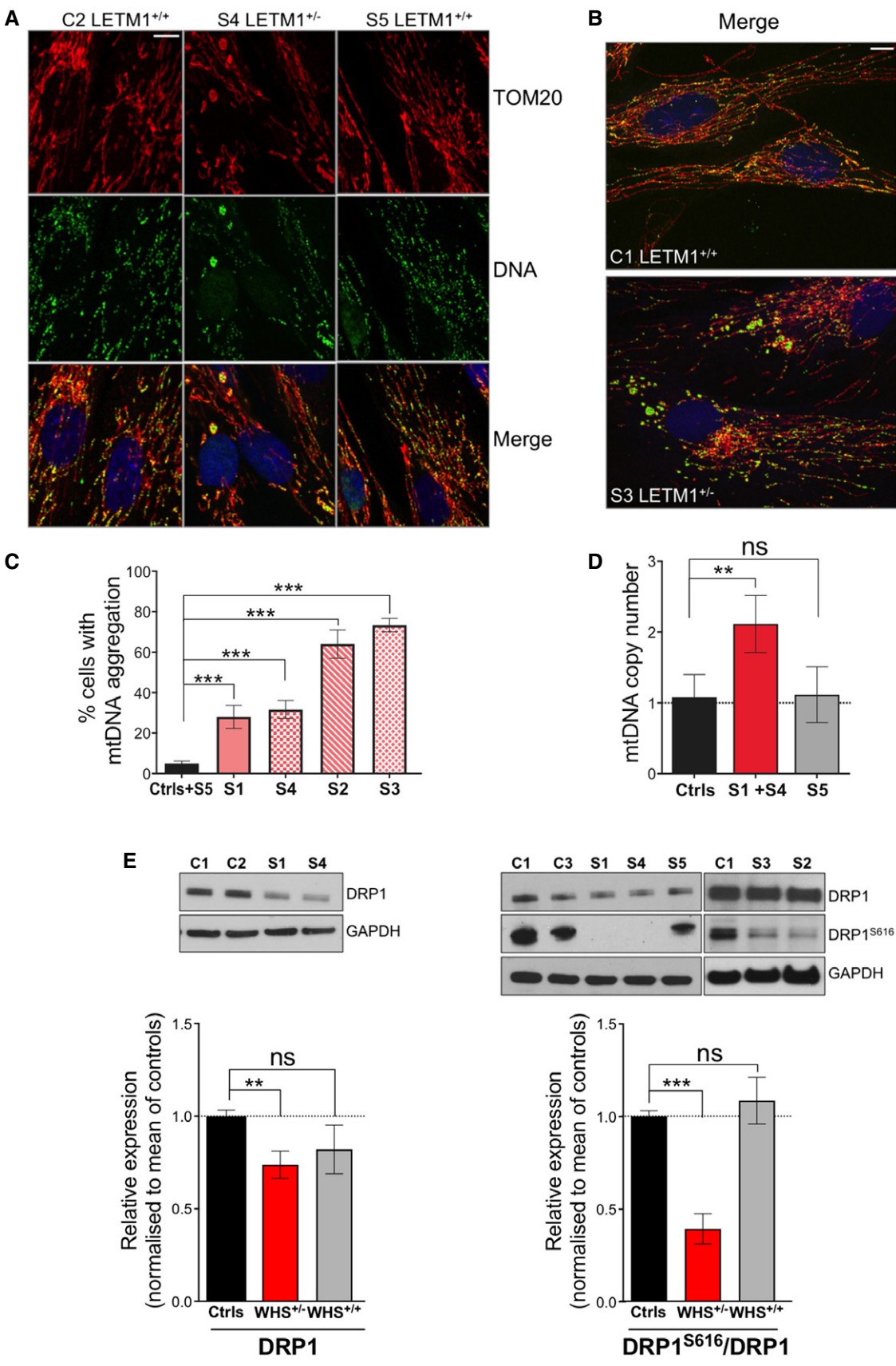

Figure 7.

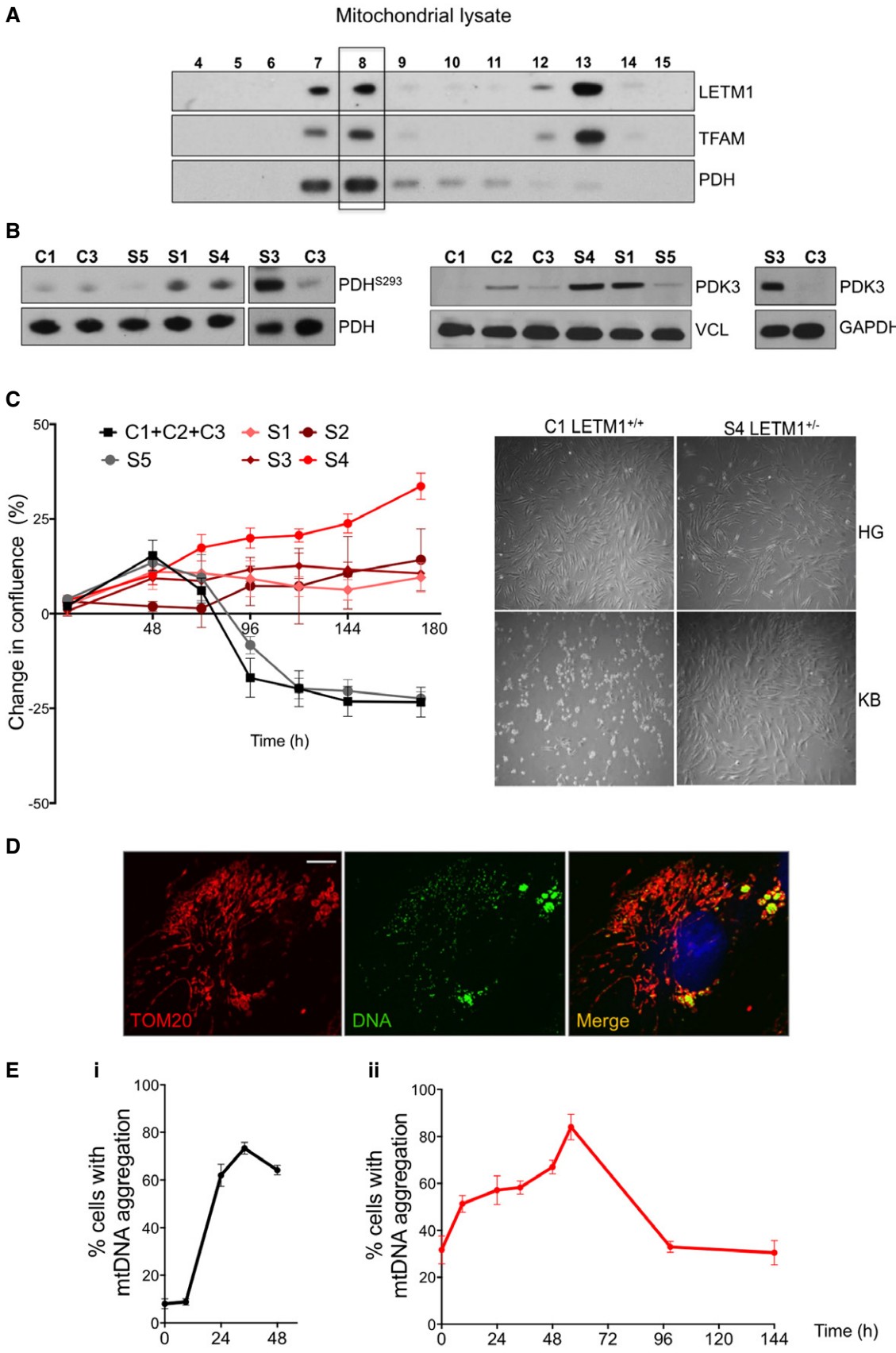

**Figure 8.**

**Figure 8. WHS LETM1$^{+/-}$ fibroblasts show reduced PDH phosphorylation and tolerate a strict ketogenic diet.**

A    Representative immunoblots of mitochondrial lysates of HEK cells separated on a 20–42.5% iodixanol gradients probed for LETM1, TFAM, and PDH.
B    Left panel: immunoblot analysis of total PDH protein levels and the inactive phosphorylated (repressed form) PDH$^{S293}$, in control (C1, C3), WHS *LETM1*$^{+/-}$ (S1, S3, S4), and WHS LETM1$^{+/+}$ (S5) fibroblasts. Center and right panels: Immunoblot analysis of PDK3 in control (C1–C3), WHS *LETM1*$^{+/-}$ (S1, S3, and S4), and WHS *LETM1*$^{+/+}$ (S5) fibroblasts.
C    Left panel: change in cell confluence of control (C1–C3), WHS LETM1$^{+/-}$ (S1-S4), and WHS *LETM1*$^{+/+}$ (S5) fibroblasts grown in BHB over the course of 188 h. Data are expressed as mean ± SEM of *n* = 3 independent experiments. Right panel: images of control and WHS *LETM1*$^{+/-}$ S4 cells growing either in the presence of 25 mM glucose (HG) or BHB (KB) for 6 days.
D    Confocal images of control cells (C1) grown for 24 h in the presence of BHB and immunolabeled with anti-TOM20 (red) and anti-DNA (green), and DAPI-stained (blue). Scale bar 6 μm.
E    Proportion of cells that displayed mtDNA aggregation during the KB treatment (panel i—control cells (C1, C2) and WHS LETM1$^{+/+}$ (S5) fibroblasts; panel ii—LETM1$^{+/-}$ (S1, S2, S4) fibroblasts). At least 50 cells per cell line were counted at the indicated time points. Data are expressed as mean ± SEM of *n* = 3 independent experiments.

Source data are available online for this figure.

C), which can be attributed to LETM1 deficiency, as phosphorylated PDH-E1 was also elevated in LETM1 repressed HeLa cells (Appendix Fig S7). Commensurate with the increased phosphorylation of PDH-E1, WHS fibroblasts had elevated PDH kinase 3 (PDK3; Figs 8B and EV3A), which targets the E1 subunit (Kolobova *et al*, 2001; Zhang *et al*, 2014). These results imply that pyruvate oxidation is low in WHS fibroblasts compared to control cells, and thus, WHS mitochondria might be configured to utilize carbon sources other than glucose, such as fatty acids or ketone bodies. In support of this interpretation, fibroblasts cultured in medium containing the ketone body β-hydroxybutyrate (BHB) in place of glucose, suffered a "metabolic crisis" that led to extensive cell death, from which the fibroblasts of four healthy controls and subject S5 failed to recover, while the cells of all four (*LETM1*$^{+/-}$) WHS subjects continued to divide even after passaging (Figs 8C and EV3B; Appendix Fig S8; Movies EV1 and EV2).

**Nutrient availability alters mtDNA organization and expression**

The changes in mtDNA organization and nutrient utilization resulting from LETM1 deficiency suggested that the two might be linked. Therefore, the effect on mtDNA distribution of replacing glucose and pyruvate with ketone bodies (KB) was assessed. The number of control fibroblasts with mtDNA aggregates increased sixfold within 24 h of switching to BHB (Fig 8D and E) and remained at this level until the cells died (Fig 8C and E-i). Replacement of glucose and pyruvate with KB further increased the number of WHS *LETM1*$^{+/-}$ fibroblasts displaying mtDNA clustering to a peak of 85% of all the cells after 56 h (Fig 8E-ii); however, this subsided close to the original level by 99 h and continued to fall thereafter, such that, after 3 weeks of the KB regime, mtDNA organization and distribution appeared similar to control cells (Appendix Fig S9). To further test the link between nutrients and mtDNA aggregation, we cultured control fibroblasts in 5 mM glucose and 0.3 mM BHB with or without 1 mM pyruvate. Neither decreasing the glucose concentration from 25 to 5 mM, nor supplementing the medium with ketone bodies markedly altered mtDNA organization, whereas the withdrawal of pyruvate induced mtDNA clustering in cells to a similar extent as glucose and pyruvate replacement with KB (Fig 9A). Moreover, the mitochondrial translation impairment induced by LETM1 deficiency is influenced by nutrients, as the replacement of glucose with KB (BHB or acetoacetate) increased the amount of newly synthesized mitochondrial protein in HeLa cells depleted of LETM1 compared to the same cells grown in 25 mM glucose and 1 mM pyruvate (Fig 9B). These findings indicate that mtDNA organization, distribution, and expression are influenced directly by nutrient availability.

**Figure 9. Mitochondrial DNA re-organization in response to nutrient changes.**

A    Proportion of mtDNA cluster following 48 or 96 h of growth in medium with (+) or without (−) 25 mM glucose (HG), 5 mM glucose (LG), β-hydroxybutyrate (BHB), or 1 mM pyruvate (pyr). At least 50 cells per cell line were counted per condition from two independent experiments. Data are expressed as mean ± SEM. Representative images appear below the charts. Scale bar: 15 μm.
B    $^{35}$S labeling of *de novo* mitochondrial protein synthesis for 1 h, 72 h after transfection of HeLa cells with either a non-target dsRNA (NT) or *LETM1* siR1; KBs (0.1 mM BHB or acetoacetate (AA)) replaced HG (25 mM glucose, 1 mM pyruvate) 9 h before the labeling reaction. Polypeptide assignments flank the gel image; Coomassie-stained gels are used as loading controls, and immunoblots indicate the efficiency of *LETM1* knockdown.
C    Perceived or actual nutrient availability alters the organization of mitochondrial nucleoprotein complexes. In yeasts, mitochondrial ribosomes are physically linked to mitochondrial nucleoids (Kehrein *et al*, 2015), and purification studies of a mitochondrial ribosomal protein (Rorbach *et al*, 2008) and mitochondrial nucleoid proteins (He *et al*, 2012a) suggest that the same is true of mammalian mitochondria, as illustrated in (a). The current study indicates that changes in nutrient availability (a switch from glucose to ketone body supplemented culture medium) lead to re-organization of mitochondrial nucleoprotein complexes (MNPCs; panel A and Fig 8D and E, and illustrated in (b)). We infer that MNPC disorganization in LETM1 deficiency is the result of a failure to utilize pyruvate—a defect in nutrient sensing—that can be attributed to repression of pyruvate dehydrogenase (PDH) via PDK3 (Figs 8B and EV3A). Because the bulk of PDH appears to be physically associated with MNPC (Fig 8A), this could be mediated by the LETM1 bound to mitochondrial ribosomes/MNPC (Fig 4C). The direct and marked impact of nutrients on MNPCs is further evidenced by the adaptation of WHS cells (*LETM1*$^{+/-}$) to KBs (Figs 8E-ii and EV3; and Appendix Figs S8 and S9), and the demonstration that KBs mitigate the mitochondrial translation impairment caused by LETM1 depletion (Fig 9B). Hence, WHS cells progressively adapt mtDNA organization to the alternative carbon source (c), whereas control cells are not primed for KB utilization, and in their case, MNPC clustering is followed by cell death (d) (Fig 8C and Movie EV1).

Source data are available online for this figure.

    

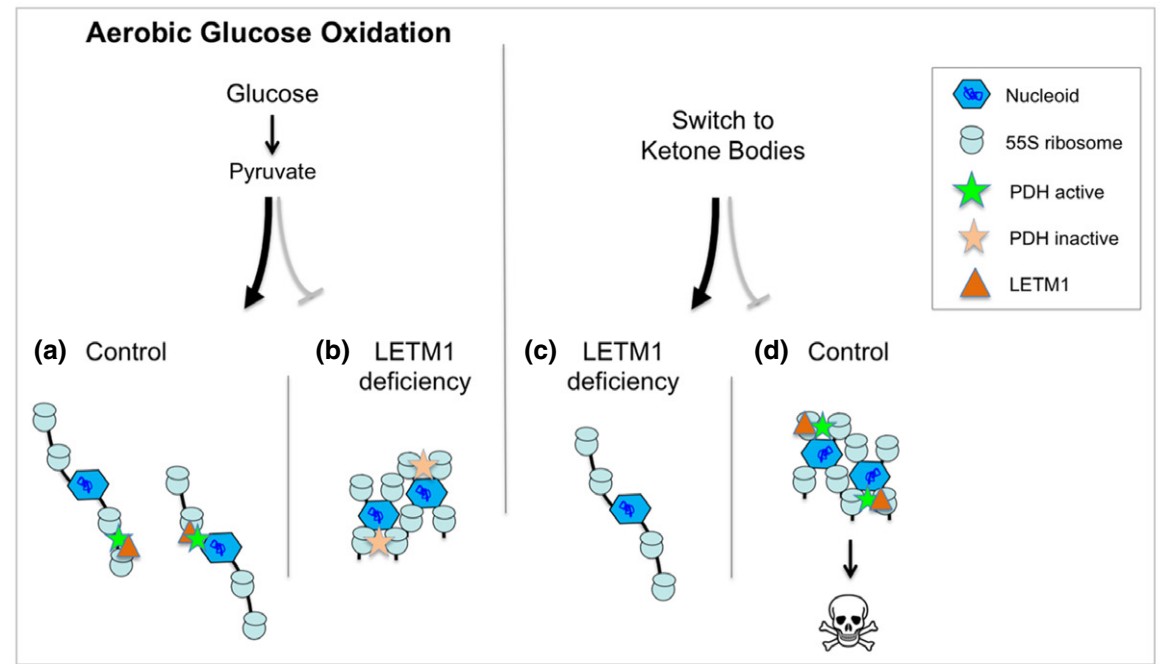

**Figure 9.**

# Discussion

There has been considerable debate as to the roles of LETM1 in mitochondrial metabolism, the resolution of which was needed to assess its possible contribution to the pathogenesis of WHS. Here, we demonstrate that mitochondrial abnormalities are evident in primary cells of WHS subjects lacking one allele of *LETM1* and that reduced LETM1 expression is associated with the remodeling of nutrient metabolism in a disease state. Although WHS deletions involve a number of other genes, the loss of which could contribute to the metabolic and mitochondrial phenotypes, the mitochondrial phenotypes were exclusive to the WHS cells lacking a *LETM1* allele and were evident in HeLa cells after *LETM1* silencing. Moreover, altered glucose and pyruvate metabolism are features of $Letm1^{+/-}$ mice, where no other gene is mutated (Jiang *et al*, 2013). Therefore, we conclude that when LETM1 is scarce, there is a switch to the consumption of ketone bodies, even in the presence of high levels of glucose, and that the decreased reliance of WHS fibroblasts on glucose and pyruvate primes them to survive growth supported on ketone bodies (Figs 8 and EV3B; Appendix Figs S8 and S9; and Movie EV2). The inference that LETM1 promotes the oxidation of glucose over lipid and amino acid catabolism has potentially important clinical implications: WHS patients could benefit from a strict diet comprising high lipid and proteins with low carbohydrate and sugars. A controlled diet of this nature may be particularly important during development (Pliss *et al*, 2016), including throughout childhood, or to control epilepsy, as has been proposed for PDH deficiency (Sofou *et al*, 2017).

### Mitochondrial DNA organization and nutrient utilization

The re-organization of the mtDNA and tolerance of a ketone body growth regime in WHS fibroblasts (Figs 8, 9 and EV3; and Appendix Figs S8 and S9), and the association of PDH with mitochondrial nucleoprotein complexes in both human cells (Fig 8A) and Xenopus oocytes (Bogenhagen *et al*, 2003), suggests that the physical interaction serves a physiological purpose. That is, proteins determining substrate utilization, such as PDH, act directly on mitochondrial nucleoids and ribosomes to couple fuel consumption and energy production. The mechanism of such coupling may well involve ion homeostasis that is known to be important for PDH activity and the TCA cycle in general (Denton *et al*, 1972; Nichols & Denton, 1995; Llorente-Folch *et al*, 2015), and in which LETM1 is heavily implicated (Nowikovsky *et al*, 2004; Jiang *et al*, 2009; Hashimi *et al*, 2013; Nowikovsky & Bernardi, 2014; Tsai *et al*, 2014). Moreover, the strict link between LETM1 and pyruvate oxidation in mitochondria and increased dependence on alternative fuels in LETM1-deficient cells may explain why there are contrasting results relating to mitochondrial dysfunction in WHS (Dimmer *et al*, 2008; Tamai *et al*, 2008, and this report).

### Mitochondrial DNA organization and energy distribution

A growing number of mitochondrial disorders are now understood to feature mtDNA aggregation or clustering (Almalki *et al*, 2014; Akman *et al*, 2016; Desai *et al*, 2017; Nicholls *et al*, 2018 and this report). Clustering of the different complexes appears to be remarkably well tolerated, in that OXPHOS subunits continue to be produced and assembled correctly in the vicinity of the mtDNA aggregates, as respiratory capacity can remain at normal levels (Appendix Fig S5). Although the failure to distribute mtDNA and its products throughout the cell need not severely compromise respiratory capacity, it can be expected to create problems in tissues such as muscle and brain, which are those most often affected in mitochondrial disorders. Muscle requires a uniform energy supply for maximal efficiency (Glancy *et al*, 2015), and to function, neurons must deliver mitochondrial cargoes to distant synapses. In the case of mtDNA aggregation and concomitant mitochondrial morphology defects, the mitochondria for transport could have a dearth of mtDNA and OXPHOS complexes, which may explain the severe neurological features in such diseases.

The different outcomes for DRP1 expression in the context of LETM1 deficiency offer further insights into LETM1's role and the architecture of the mtDNA in the organelles. LETM1 deficiency is associated with mitochondrial distensions, but these are mostly within the reticular network when DRP1 activity is co-repressed (LETM1 siR3 and occasionally siR2), whereas when LETM1 is scarce and DRP1 activity is not repressed, swollen mitochondria are ubiquitous and the network is fragmented (siR1; Fig 1A). Although the latter arrangement maintains a near-normal mtDNA distribution in the face of LETM1 deficiency (Figs 3A and EV1A—siR1), clustering of the macromolecular complexes might be preferable, to help to counteract the hypotonic environment, and this could explain the frequent spontaneous co-repression of DRP1 (Figs 5A and 7E), which evidently improves mitochondrial translation and OXPHOS when LETM1 is depleted (Fig 5B and C; and Appendix Fig S3B). Given that the findings suggest that repressing DRP1 attenuates the mitochondrial dysfunction caused by LETM1 deficiency, it will be important to determine whether some cell types of WHS $LETM1^{+/-}$ patients maintain normal levels of DRP1, as, if so, decreasing DRP1 might improve their symptoms or slow disease progression.

Mitochondria are known to adapt physically to nutrient availability (Gomes *et al*, 2011; Rambold *et al*, 2011). The current study demonstrates that changes in nutrient availability and utilization remodel the nucleoprotein complexes in mitochondria and thereby indicates how nutrients can modulate gene expression and energy production in the organelle (Fig 9C). An important corollary is that genetic defects in metabolic factors linked to mitochondrial nucleoprotein complexes, or their regulators, can produce a pseudo-starvation state, owing to an inability to utilize an available nutrient.

# Materials and Methods

### Human studies

Informed consent was obtained for all the subjects included in the study, and the procedures followed were in accordance with the WMA Declaration of Helsinki and the Department of Health and Human Services Belmont Report.

### Cell lines and culture conditions

Primary skin fibroblasts were obtained from healthy controls ($n = 4$) and from Wolf–Hirschhorn syndrome (WHS) patients ($n = 5$; Appendix Table S1). Unless stated otherwise, fibroblasts,

HeLa, and human embryonic kidney cells (HEK293T, Invitrogen) were grown in Dulbecco's modified Eagle's medium (DMEM, Life Technologies) supplemented with 25 mM glucose, 1 mM pyruvate, 10% fetal bovine serum (FBS, Pan Biotech UK), and 1% penicillin and streptomycin (PS, Life Technologies) at 37°C in a 5% $CO_2$ atmosphere. All cells were tested for mycoplasma infection on regular basis using LookOut Mycoplasma PCR Detection Kit (Sigma).

## RNA interference of *LETM1* and *DRP1* in HeLa cells

For knockdown experiments, HeLa cells were transfected using either siRNAs targeting *LETM1* (Appendix Table S2 and Appendix Fig S1A) or no particular gene (NT). All oligos were transfected at a concentration of 10 nM using Lipofectamine RNAiMAX (Life Technologies), according to the manufacturer's specifications. In general, HeLa cells were analyzed after 6 days, following two rounds of siRNA with the second transfection performed 3 days after the first. Where indicated the exceptions were 72 h after one round of transfection. Simultaneous knockdown of *LETM1* and *DRP1* was performed as described above using the siRNA targeting *LETM1* (siR1) and *DRP1* (Appendix Table S2).

## Cell lysis, SDS–PAGE, Western blotting, and immunodetection

Cells were lysed in PBS, 0.1% n-dodecyl-D-maltoside (DDM, Sigma), 1% SDS, 50 U benzonase (Novagen), 1:50 (v/v) protease inhibitor cocktail (Roche), and 1:100 (v/v) phosphatase inhibitor (Cell Signaling), and protein concentration was measured by Lowry assay (Bio-Rad). 10–20 μg of proteins was separated by SDS–PAGE on Bis-Tris NuPAGE gels (Life Technologies) and transferred to polyvinylidene fluoride membrane (Millipore). Membranes were blocked in 5% non-fat dry milk in PBS with 0.1% (v/v) Tween-20 (PBST) for 1 h at RT (room temperature) and then incubated overnight with primary antibodies (Appendix Table S3) at 4°C. The day after, membranes were washed three times with PBST and incubated for 1 h at RT with the appropriate secondary antibodies. Membranes were then washed three times with PBST, and signals were developed using ECL (GE Healthcare).

## [35S]-Methionine in cell labeling for mitochondrial protein synthesis

Mitochondrial translation products were labeled using [35]S-methionine as described previously (Dalla Rosa *et al*, 2014) with some modifications. Fibroblasts or HeLa cells were washed twice with methionine-/cysteine-free DMEM (Life Technologies) supplemented with 1 mM L-GlutaMAX, 96 μg/ml cysteine (Sigma), 1 mM pyruvate, and 5% (v/v) dialyzed FBS, and incubated in the same medium for 10 min at 37°C. 100 μg/ml emetine dihydrochloride (Sigma) was then added to inhibit cytosolic translation, before pulse-labeling with 100 μCi [35S]-methionine for 45–60 min. Cell were chased for 10 min at 37°C in regular DMEM with 10% FBS, washed three times with PBS, and collected. Labeled cells were lysed in PBS, 0.1% n-dodecyl-D-maltoside (DDM), 1% SDS, 50 units of benzonase (Novagen), and 1:50 (v/v) Roche protease inhibitor cocktail. Protein concentration was measured by Lowry assay, and 20 μg of protein was separated by 12% SDS–PAGE. Gels were dried and exposed to phosphor plates, and the signal was detected by PhosporImager.

## Cellular oxygen consumption

The oxygen consumption rate (OCR) of HeLa cells was assayed with a XF24 Extracellular Flux Analyzer (Agilent). Briefly, $4 \times 10^4$ proliferating cells were seeded in microplates (Agilent) in 250 μl of prewarmed growth medium (DMEM, Gibco) and incubated 37°C/5% $CO_2$ for 8 h. Subsequently, the medium was removed and replaced with assay medium (XF Base minimal DMEM (Agilent) complemented with 2 mM glucose, 2 mM GlutaMAX, and 1 mM pyruvate) and cells were incubated for 30 min in a 37°C non-$CO_2$ incubator. After taking an OCR baseline measurement, 1 μM oligomycin, 1.2 μM carbonyl cyanide-4-trifluoromethoxyphenylhydrazone (FCCP), and 1 μM rotenone were added sequentially. For normalization, protein concentration in each well was determined using the Lowry assay (Bio-Rad).

## mtDNA copy number

Quantification of mtDNA was performed as described in Dalla Rosa *et al* (2016) with some modifications. Briefly, total DNA was isolated from frozen pellets of human fibroblasts or HeLa cells using DNeasy Blood and Tissues Kit (Qiagen), according to the manufacturer's protocol. Real-time quantitative PCR was performed in triplicate on 384-Well Reaction Plates (Applied Biosystems), in final volumes of 10 μl. Each reaction contained 20 ng DNA template, 1× Power SYBR Green PCR Master Mix (Applied Biosystems), and 0.5 μM of forward and reverse primers. Mitochondrial and nuclear DNAs were amplified using primers specific to regions of human COXII and APP1 genes, respectively. The sequences of the primers used are in Appendix Table S2. Changes in mtDNA amount were calculated using the $2^{-\Delta\Delta Ct}$ method (Schmittgen & Livak, 2008) and represented as fold change relative to the indicated control.

## RNA extraction and analysis of mitochondrial transcripts by qPCR

For total RNA extraction, HeLa cells were lysed in TRIzol reagent (Life Technologies) and RNA, DNA, and proteins were separated by the addition of chloroform. After centrifugation (15 min, 12,000 g at 4°C), the upper aqueous phase containing RNA was transferred to a fresh tube and precipitated with equal volume of 70% ethanol. This solution was then loaded onto a PureLink RNA Mini Kit (Invitrogen) column, and RNA extraction was completed as per the manufacturer's instruction. Relative transcript abundance was measured by qPCR on a 7900HT Fast Real-Time PCR System (Applied Biosystems). Transcript levels were normalized to housekeeping transcript β-2-globulin. Primers are detailed in Appendix Table S2.

## Mitochondrial isolation, iodixanol, and sucrose gradient fractionation

Mitochondria were isolated as previously described (Dalla Rosa *et al*, 2014). Briefly, HEK293T cells were disrupted by homogenization in hypotonic buffer (20 mM HEPES pH 8, 5 mM KCl, 1.5 mM $MgCl_2$, and 2 mM DTT) and mixed with a mannitol–sucrose buffer to final concentration of 210 mM mannitol, 70 mM sucrose, 20 mM HEPES pH 8, and 2 mM EDTA (1× MSH), prior to purification of mitochondria by differential centrifugation. For iodixanol gradient fractionation, 1,000 g supernatant from lysed mitochondria was

loaded onto 20–42.5% discontinuous iodixanol (OptiPrep, Sigma) gradients (Dalla Rosa *et al*, 2014) and centrifuged at 100,000 *g* for 14 h at 4°C. Resulting gradients were fractionated into 0.5 ml fractions collected from the bottom of the tube and subjected to Western blot analysis. For the analysis of the mitochondrial ribosome, mitochondria were isolated as above using EDTA-free buffers (Dalla Rosa *et al*, 2014) and subsequently lysed for 20 min on ice. Two different lysis buffers used were as follows: the standard buffer containing 260 mM sucrose, 100 mM NaCl, 20 mM MgCl$_2$, 10 mM Tris–HCl pH 7.5, 1% Triton X-100, 1× protease inhibitor w/o EDTA (Roche), 0.08 U/μl rRNasin (Promega) in DEPC-treated water and a second buffer containing a lower concentration of salt (50 mM NaCl). Mitochondrial lysates were cleared by centrifugation (10,000 *g* for 45 min at 4°C) and loaded on 10–30% linear sucrose gradients made in the appropriate lysis buffer. After centrifugation at 71,000 *g* for 15 h at 4°C, 15 fractions of 750 μl each were collected from the top of the gradient using an automated gradient harvester (Brandel) and analyzed by Western blotting. In the case of *LETM1* silencing, the isokinetic sucrose gradient analysis of mitochondrial ribosomes was performed on total lysates.

### Immunofluorescence and imaging

Fibroblasts or HeLa cells grown on coverslips were fixed with 4% paraformaldehyde, permeabilized with 0.3% Triton X-100, and incubated with indicated primary antibodies (Appendix Table S4) overnight at 4°C. For bromouridine (BrU) labeling, cells grown on coverslips were incubated with 5 mM BrU for 60 min prior to fixation. Anti-BrdU antibody (Appendix Table S4) was used to detect BrU-labeled RNA. Following washes, cells were incubated for 1 h at RT with secondary antibody after which coverslips were mounted on glass slides over ProLong® Gold Antifade Reagent, which includes 4′,6-diamidino-2-phenylindole (DAPI). Samples were imaged either on a SP5 TCS Inverted Confocal Microscope (Leica Biosystem) using a 63× oil immersion objective with a numerical aperture of 1.4 (63×/1.4 Oil) or a 100× oil immersion objective with a numerical aperture of 1.46 (100×/1.46 Oil) or on Nikon Ti Inverted Confocal Microscope using 60× immersion objective. Z stack of red, green, and blue images using a step size of either 0.3 or 0.125 μm was acquired sequentially and merged using ImageJ. Confocal images were adjusted for brightness/contrast in ImageJ. Images in each figure were processed equally.

### Electron microscopy

WHS and control fibroblast cells were fixed in 2% glutaraldehyde/ 2% paraformaldehyde for 30 min, and then 1% osmium tetroxide for 1 h, using 0.1 M sodium cacodylate buffer pH 7.2, at RT. The samples were then dehydrated and embedded in Epon resin. Sections were stained with ethanolic uranyl acetate and Reynold's lead citrate, and viewed with a JEOL 100EX transmission electron microscopy (TEM). Images were captured with a Gatan Orius 1000 charge-coupled devices (CCD).

### Ketone body supplementation

Fibroblasts were cultured to reach 40–50% confluency, washed once with PBS (Life Technologies), and incubated with DMEM with

**The paper explained**

**Problem**

The mitochondrial inner membrane protein LETM1 is involved in regulating ion homeostasis and mitochondrial volume. Yet, considerable uncertainty remains around its role in organelle function and cellular metabolism, and it is not known whether mitochondrial dysfunction contributes to the pathology in the many cases of Wolf–Hirschhorn syndrome (WHS) where LETM1 is deficient.

**Results**

Here, we show that LETM1 is a component of mitochondrial nucleoprotein complexes that supports mitochondrial translation. The loss of one copy of LETM1 produces aberrant mitochondrial morphology and DNA disorganization in WHS *LETM1*$^{+/-}$ fibroblasts, phenocopying *LETM1* gene silencing. The effect on mtDNA is associated with remodeling of nutrient metabolism toward ketone body utilization.

**Impact**

The findings establish roles for LETM1 in mitochondrial gene expression and mtDNA organization and show that LETM1 insufficiency causes mitochondrial abnormalities in WHS. Moreover, the results show that the arrangement of mitochondrial nucleoids and ribosomes is tied to nutrient availability and if incorrectly configured, nutrient usage by the mitochondria is no longer matched to availability, which can result in impaired organelle function and disease.

glutamine, w/o glucose and w/o pyruvate (Life Technologies) supplemented with 0.3 mM β-hydroxybutyrate (BHB, Cayman Chemical), 10% dialyzed FBS, and 1% PS. The medium was changed every 48 h. Cell density was determined using an IncuCyte Live Cell Imaging System (Essen Instruments), and phase-contrast images were acquired at 3-h intervals over a period of 3 weeks and processed automatically.

### Statistics

All statistical calculations were performed using GraphPad Prism 7.0. Analysis of mitochondrial protein synthesis and immunoblots was performed using Fiji ImageJ. The non-parametric tests used were one-way ANOVA or *t*-test with Welch's correction. The former was used to compare more than two independent groups, while the latter was used to compare two independent groups. All error bars are ± standard error of the mean, which is an estimate of the true variation of the samples extrapolated from the standard deviation of the measured values. *P*-values < 0.05 were considered to be statistically significant and labeled as follows: \*$P < 0.05$, \*\*$P < 0.01$, and \*\*\*$P < 0.001$; ns, not statistically significant. Exact *P*-values are shown in Appendix Table S5.

**Expanded View** for this article is available online.

### Acknowledgements

The study was funded by a UK Medical Research Council senior non-clinical fellowship to AS (MC_PC_13029) and by the European Commission (MEET project grant 317433) to AS and IJH. AM and HH are supported by the UK Medical Research Council. SL, PD, GM, and MZ are supported by the Catholic University of Rome. IJH is supported by the Ikerbasque Science Foundation, The Carlos III Health Program, and Biodonostia Research Institute. We thank

all the families with the Wolf–Hirschhorn syndrome and the Italian Association for WHS. We would like to thank Professor Wiesner for TFAM antibody.

## Author contributions

Conceptualization: AS; Methodology: AS and RD; Investigation: RD, ALM, AWEJ, IDR, AM, MM, EMAH, and AS; Writing: AS and IJH; Funding Acquisition: AS; and Resources: HH, GM, SL, PND, and MZ.

## Conflict of interest

The authors declare that they have no conflict of interest.

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
