## [Review Process File · EMBO Molecular Medicine]

LETM1 couples mitochondrial DNA metabolism and nutrient preference

Romina Durigon, Alice L. Mitchell Aleck W. E. Jones, Andreea Manole, Mara Mennuni, Elizabeth M.A. Hirst, Henry Houlden, Giuseppe Maragni, Serena Lattante, Paolo Niccolo' Doronzio, Ilaria Dalla Rosa, Marcella Zollino, Ian J. Holt & Antonella Spinazzola

Review timeline:

Submission date:	02 October 2017
Editorial Decision:	14 November 2017
Revision received:	04 May 2018
Editorial Decision:	28 May 2018
Revision received:	07 June 2018
Accepted:	14 June 2018

Editor: Céline Carret

Transaction Report:

1st Editorial Decision

14 November 2017

Thank you for the submission of your manuscript to EMBO Molecular Medicine. We have now heard back from the three referees whom we asked to evaluate your manuscript.

As you will see from the reports below, the referees find the topic of your study of interest. However, they raise substantial concerns on your work, pertaining particularly to over-interpretations, limited mechanistic understandings, and suboptimal use of cellular models. All three referees provide a long and detailed list of suggestions for improving the study.

Overall it is clear that publication of the manuscript cannot be considered at this stage. I also note that addressing the reviewers concerns in full will be necessary for further considering the manuscript in our journal and this appears to require a lot of additional work and experimentation. I am unsure whether you will be able or willing to address those and return a revised manuscript within the 3 months deadline. On the other hand, given the potential interest of the findings, I would be willing to consider a revised manuscript with the understanding that the referees' concerns must be fully addressed and that acceptance of the manuscript would entail a second round of review.

Please note that EMBO Molecular Medicine strongly supports a single round of revision and that, as acceptance or rejection of the manuscript will depend on another round of review, your responses should be as complete as possible.

I look forward to receiving your revised manuscript.

Should you find that the requested revisions are not feasible within the constraints outlined here and choose, therefore, to submit your paper elsewhere, we would welcome a message to this effect.

***** Reviewer's comments *****

Referee #1 (Remarks for Author):

Thank you very much for sending me this manuscript from Durigon et al to review. As well described by the authors, there has been a great deal of speculation regarding LETM1 and its role in mitochondrial homeostasis, with several conflicting reports. It is not simple, but it would be terrific if something could be published that reconciled these conflicts. In this paper, I think there is rather too much speculation and extrapolation in general, but I was intrigued by the data on the use of ketone bodies as fuel for the LETM1 heterozygotes and this data would make EMM a reasonable target journal for publication. I have comments and questions, below.

I am rather confused by the few concluding sentences on p.6 before the section on mitochondrial swelling. In my figures, there is little evidence to suggest the 28S subunit markers are more depleted in the LETM1 depletion expts than the 39S markers. Further, in Fig 3C siR1 actually results in slightly more depletion of 16S than in 12S. Therefore it is odd to suggest that LETM1 is involved in assembly of the 28S. Available data suggests stability and assembly of the 39S is unaffected when 28S assembly is impaired. If LETM1 were involved in 28S, why would the markers of the 39S equally deplete? I think it is also a big stretch to implicate LETM1 in facilitating RNA delivery to the 28S. The final sentence is even more of a stretch. The authors claim that the transcripts remain associated with the nucleoids in the absence of LETM1 and that this would impede further rounds of transcription. Very interesting if this was true, but surely this is a claim that is unsubstantiated by any data?

In the discussion, p.11 mtDNA organization, I don't understand what the authors mean by 'remodelling of the mtDNA to a configuration appropriate to ketone body usage?

p.12, the second sentence of the section on mtDNA organisation and energy distribution is a non-sequitor from the first sentence. I don't understand the why the claim that mtDNA determines the distribution of ribosomes/OXPHOS complexes (if this is really what is occurring) explains why many mitochondrial defects result in mtDNA aggregation?

What was the final concentration of BHB used? There is no indication of whether the DMEM used was supplemented with glutamine or non-essential amino acids.

A general point. Are these true mtDNA aggregates that are being seen, or are they indications of mtDNA accumulation? For example, if true aggregates then one might expect to see evidence of this in the pellet or heavier fractions in the iodixonal gradient. The reason being so pedantic about this is that aggregation may cause loss of mtDNA transcription or replication.

In summary, this is an interesting paper with a lot of work, which gives more insight into how cells respond to LETM1 deficiency. Further, it shows for the first time that there are some mitochondrial structural abnormalities in LETM1 deficient patient cell lines that do not manifest in these lines as defects in OXPHOS and related mtDNA expression defects. There is an intriguing observation that growth in ketone bodies is tolerated much more freely than control cell lines. This may have some relevance to treatment for patients with LETM1 deficiency but we are some way off from showing this. It is still unclear exactly what the molecular function of LETM1 could be.

Referee #2 (Comments on Novelty/Model System for Author):

Two models are used. HeLa cells are sufficient for mechanistic studies on LETM1 function. More pathophysiological studies are performed in fibroblasts derived from HWS patients. This study provides new connections between mitochondrial gene expression and respiratory chain substrate availability and utilization.

Referee #2 (Remarks for Author):

This manuscript by Durigon and colleagues present novel information regarding the functions of mitochondrial protein LETM1 in health and in the context of Wolf-Hirschhorn syndrome (WHS). The authors show that LETM1 interact with the mitochondrial nucleoids thus affecting mitochondrial gene expression and the activity of nucleoid-associated enzymes, most prominently, pyruvate dehydrogenase. These data, obtained in fibroblasts from WT and WHS patients as well as HeLa cells silenced for LETM1, establish a connection between mitochondrial DNA expression and nutrient utilization that in the case of WHS is manifested as mitochondrial dysfunction. The authors show that a modified lipid-rich and carbohydrate-poor diet may be effective to mitigate the effect of the LETM1 haploinsufficiency typical of WHS.

The manuscript is technically sound and the results presented, particularly connecting LETM1 function to respiratory chain substrate preference are extremely interesting. However, the connections between nucleoids and the mitochondrial protein synthesis machinery should be explored further

1- LETM1 is proposed to act on the assembly of the mtSSU. However, its yeast homolog seems to participate in the tethering of the mtLSU to the membrane. Do the authors believe that LETM1 is playing more than one role in mitochondrial ribosome assembly?

2- In Fig. 1, one wonders how much of the LETM1-silencing effect on mitochondrial protein synthesis efficiency relates to lowered levels of mitochondrial ribosomes or to fast degradation of newly synthesized proteins that fail to be inserted into the membrane in the absence of LETM1 (as it occurs in yeast). The protein synthesis experiment should include some shorter pulse times followed by chase if necessary.

3- In Fig 3A, ribosomal proteins are shown to co-localize with nucleoid proteins and LETM1. However, this co-localization may not explain the whole story. Questions remaining are:

2.1- is LETM1 interacting with actively transcribing nucleoids?

2.2- Does LETM1 interacts (or is part of) the mitochondrial nucleoids and the mitochondrial RNA granules too? In fig 4, for example, it would be of interest to stain for RNA.

2.3- In Fig 10, all ribosomes are depicted interacting with the nucleoids; however, the manuscript does not present enough data to support that possibility.

4- What is RETC standing for in Fig 10?

5- I would suggest the authors to use the new nomenclature for ribosomal proteins that was recently established (Ban et al)

Referee #3 (Comments on Novelty/Model System for Author):

The HeLa model with transient silencing of LETM1 is difficult to interpret. Stable silencing models with clear gene dosage effects would improve the study.

Referee #3 (Remarks for Author):

The manuscript by Durigon and colleagues investigates the effects of LETM1 downregulation on mitochondria. LETM1 is one of the chromosome 4 genes that can be deleted in WHS. Therefore, understanding the mitochondrial consequences of LETM1 deficiency may have implications for disease management. LETM1 is thought to play a role in ion regulation in the mitochondrial matrix, but this work potentially highlights different functions. Using both HeLa cells in which the LETM1 gene is partially silenced or primary fibroblasts from WHS patients, the authors show several mitochondrial abnormalities, which include mtDNA-encoded protein expression, mitochondrial nucleoids distribution, mitochondrial fission, and metabolic regulation. The latter is likely the most disease relevant observation, as the mouse model of heterozygote LETM1 KO suffers from glucose utilization problems and severe neurological impairments.

The results are very interesting and strongly suggest that LETM1 may have roles far more complex than originally described. However, they are also quite preliminary and correlative in nature. At this

stage, it is difficult to establish which of the many diverse consequences of LETM1 downregulation described in the different cellular systems are directly and specifically associated with the proposed protein functions and which are secondary. Furthermore, there are several concerns on methodology and data presentation that weaken the conclusions.

Specific points

- 1) Cell culture systems: The relevance of the siRNA experiments in HeLa cells is questionable. The acute siRNA transfection appears to result in inconsistent protein silencing across siRNA and experiments. It is completely unclear how the different results are selected for presentation and result interpretation. At times, the same siRNA appears to silence LETM1 almost completely and at other times less than 50%. The issue of the time points is also a concern. How certain time points are selected and why is unclear. The primary cells are much more reliable in terms of reproducing the genetics of WHS. However, there is line to line variability, as expected. The authors present results in some of the four lines available in a seemingly random manner. Sometimes Lines 1 and 4 are shown and other times line 3. It is really difficult to determine if the results shown are truly representative of the syndrome as a whole. The study would really benefit of the use of a much better-defined cell system, such as stable cell lines with homo and heterozygote LETM1 KO (e.g., CRISP or TALEN). If the homozygote KO is lethal in cells, as it is in the mouse, this should be at least known. For the fibroblasts, at a minimum, results from all available cell lines should be shown. Ideally, the phenotypes in both silenced HeLa and fibroblasts should be suppressed by expression of exogenous LETM1.
- 2) The findings described in HeLa cells with severe knockdown of LETM1 appear quite variable depending on which phenotype is analyzed. Overall, there are several diverse consequences, each interesting in itself. However, it is unclear how they relate to each other and how consistent they are. The relationship between mitochondrial morphology and nucleoid distribution, for example, could be one of cause and effect, but which comes first is unknown. The reader is left trying to connect the dots, a daunting task. The alterations in PDH phosphorylation are presented at the end of the results, but it seems that the authors think that they may be upstream of the changes in DRP1 activation, mitochondrial morphology, and nucleoid distribution. If this is the logical thread, then the metabolic reshaping should be the centerpiece of the work, also in view of the mouse results.
- 3) The study of PDH regulation is underdeveloped and not much new information is revealed, beyond the data that were published by others on the heterozygote mouse. For example, it would be interesting to show what would happen if PDH was reactivated by treatment with PDK inhibitors, such as dichloroacetate. In addition, PDH activity should be measured in WHS fibroblasts and in HeLa cells with LETM1 silencing.
- 4) The expression of the KB utilization pathway is not investigated. Therefore, it is not exactly known how WHS cells adapt to decreased PDH activity.
- 5) If the hypothesis is correct, lactate production should be increased in HeLa cells silenced and in WHS fibroblasts.
- 6) As fibroblasts with 50% LETM1 do not have mtDNA-encoded protein translation defects or oxidative phosphorylation defects, while HeLa cells with less than 50% do, it is unclear whether LETM1 has two or more independent functions in mitochondria, involving PDH regulation, mitochondrial protein translation, and ion homeostasis. Are these functions related?
- 7) LETM1 is involved in Ca²⁺ and K⁺ exchange across the inner membrane. It would be important to know if these functions are in any way associated with the phenotypes described here.
- 8) Several results, especially the cell imaging and the western blots of the fractions, are not quantified. This is important for the dataset for which the authors make statements on significant differences.

General:

In light of the reviewers' comments, questions and recommendations, we have carried out a substantial amount of additional work. The revised manuscript includes more data and reproduces the earlier results in additional WHS cell lines to show the findings are robust. We have also reduced the amount of speculation, or provided further support for the conclusions and inferences. We have carried out additional experiments that indicate LETM1 is associated with mitochondrial ribosomes, as opposed to mtDNA or newly synthesized RNA. This was helpful for us and makes it

more straightforward for the reader, as some uncertainty about links to nucleoids or ribosomes has been removed.

Furthermore, we have rewritten the manuscript in an effort to make it more accessible to the reader; provided the missing quantification; and extended the analysis of nutrient availability in three ways. In new experiments, we show that ketone bodies mitigate the adverse effects of LETM1 deficiency on mitochondrial DNA organization and protein synthesis, and that the withdrawal of 1 mM pyruvate from the culture medium induces mtDNA aggregation. The new findings strongly suggest that the effects of LETM1 deficiency on mtDNA metabolism are linked to glucose/pyruvate utilization; and strengthen the original conclusion that mtDNA organization and expression are subject to regulation according to real or perceived changes in nutrient availability.

Referee #1 (Remarks for Author):

Thank you very much for sending me this manuscript from Durigon et al to review. As well described by the authors, there has been a great deal of speculation regarding LETM1 and its role in mitochondrial homeostasis, with several conflicting reports. It is not simple, but it would be terrific if something could be published that reconciled these conflicts. In this paper, I think there is rather too much speculation and extrapolation in general, but I was intrigued by the data on the use of ketone bodies as fuel for the LETM1 heterozygotes and this data would make EMM a reasonable target journal for publication. I have comments and questions, below.

I am rather confused by the few concluding sentences on p.6 before the section on mitochondrial swelling. In my figures, there is little evidence to suggest the 28S subunit markers are more depleted in the LETM1 depletion expts than the 39S markers. Further, in Fig 3C siR1 actually results in slightly more depletion of 16S than in 12S. Therefore it is odd to suggest that LETM1 is involved in assembly of the 28S. Available data suggests stability and assembly of the 39S is unaffected when 28S assembly is impaired. If LETM1 were involved in 28S, why would the markers of the 39S equally deplete? I think it is also a big stretch to implicate LETM1 in facilitating RNA delivery to the 28S.

The final sentence is even more of a stretch. The authors claim that the transcripts remain associated with the nucleoids in the absence of LETM1 and that this would impede further rounds of transcription. Very interesting if this was true, but surely this is a claim that is unsubstantiated by any data?

Having considered carefully these remarks, together with those of the other 2 reviewers, we have revised the section on the 55S ribosome. We accept that we have a limited amount of data relating to the specific issue of whether the small or the large subunit is more affected by LETM1 deficiency (notwithstanding that an error was made in the labeling of Fig. 3C, which erroneously suggested the opposite, see below). The more circumspect revised comments are also shorter and aim to make the task of following the text less daunting for the reader.

We mislabeled 16S and 12S in Fig. 3C at a late stage of assembly of the manuscript and only realized our mistake when reviewer 1 noted that the result according to the figure was the opposite of the (true) one described in the text. The figure labeling has been corrected. Moreover, these data now form part of figure 2, with the straightforward conclusion being that LETM1 deficiency leads to decreased levels of protein and RNA components of mitochondrial ribosomes.

In the discussion, p.11 mtDNA organization, I don't understand what the authors mean by 'remodelling of the mtDNA to a configuration appropriate to ketone body usage?'

To reduce the amount of speculation this sentence has been removed.

p.12, the second sentence of the section on mtDNA organisation and energy distribution is a non-sequitor from the first sentence. I don't understand the why the claim that mtDNA determines the distribution of ribosomes/OXPHOS complexes (if this is really what is occurring) explains why many mitochondrial defects result in mtDNA aggregation?

The text has been revised, including removal of the first sentence, thereby further reducing the amount of speculation.

What was the final concentration of BHB used? There is no indication of whether the DMEM used was supplemented with glutamine or non-essential amino acids.

For all the experiments on fibroblasts in the manuscript we used 0.3 mM BHB, and the DMEM plus BHB contained neither glucose nor pyruvate but was supplemented with glutamine, as stated in the revised methods.

A general point. Are these true mtDNA aggregates that are being seen, or are they indications of mtDNA accumulation? For example, if true aggregates then one might expect to see evidence of this in the pellet or heavier fractions in the iodixonal gradient. The reason being so pedantic about this is that aggregation may cause loss of mtDNA transcription or replication.

We have not seen any evidence of aggregates that are fused such that they alter the buoyant density of mitochondrial ribosomes or nucleoids. To note: 1) mtDNA copy number was increased rather than decreased in the context of LETM1 deficiency. 2) The ribosome analysis on sucrose gradients showed that the amount of 55S ribosome was decreased and more material was detected at the top end of the gradient (the opposite end from any pellet). 3. Decreases in MRPs and rRNAs were based on whole cell lysates that were fully denatured. Therefore, there is no suggestion any of the data were distorted owing to material lost in the form of aggregates forming a pellet during ultracentrifugation.

In summary, this is an interesting paper with a lot of work, which gives more insight into how cells respond to LETM1 deficiency. Further, it shows for the first time that there are some mitochondrial structural abnormalities in LETM1 deficient patient cell lines that do not manifest in these lines as defects in OXPHOS and related mtDNA expression defects. There is an intriguing observation that growth in ketone bodies is tolerated much more freely than control cell lines. This may have some relevance to treatment for patients with LETM1 deficiency but we are some way off from showing this. It is still unclear exactly what the molecular function of LETM1 could be.

Referee #2 (Comments on Novelty/Model System for Author):

Two models are used. HeLa cells are sufficient for mechanistic studies on LETM1 function. More pathophysiological studies are performed in fibroblasts derived from HWS patients. This study provides new connections between mitochondrial gene expression and respiratory chain substrate availability and utilization.

Referee #2 (Remarks for Author):

This manuscript by Durigon and colleagues present novel information regarding the functions of mitochondrial protein LETM1 in health and in the context of Wolf-Hirschhorn syndrome (WHS). The authors show that LETM1 interact with the mitochondrial nucleoids thus affecting mitochondrial gene expression and the activity of nucleoid-associated enzymes, most prominently, pyruvate dehydrogenase. These data, obtained in fibroblasts from WT and WHS patients as well as HeLa cells silenced for LETM1, establish a connection between mitochondrial DNA expression and nutrient utilization that in the case of WHS is manifested as mitochondrial dysfunction. The authors show that a modified lipidrich and carbohydrate-poor diet may be effective to mitigate the effect of the LETM1 haploinsufficiency typical of WHS.

The manuscript is technically sound and the results presented, particularly connecting LETM1 function to respiratory chain substrate preference are extremely interesting. However, the connections between nucleoids and the mitochondrial protein synthesis machinery should be explored further

1- LETM1 is proposed to act on the assembly of the mtSSU. However, its yeast homolog seems to participate in the tethering of the mtLSU to the membrane. Do the authors believe that LETM1 is playing more than one role in mitochondrial ribosome assembly?

As indicated above in the response to reviewer 1, we accept that the existing data do not suggest that LETM1 primarily affects the small rather than the large ribosomal subunit, and we have revised the text accordingly. We doubt that a protein that readily dissociates from 55S ribosomes (Fig 6) serves as an anchor, and think it more likely that it modulates ion homeostasis to facilitate translation.

As indicated in the revised text, the data fit the model for yeast ribosomes being linked to nucleoids, as this can account for the mtDNA and mtRNA abnormalities, as well as the ribosome maintenance defects. The concluding section on the interaction of LETM1 with the 55S ribosome now reads:

“These findings indicate LETM1 physically interacts with the human mitochondrial ribosome, which can include interaction with mRNA (activators), as has been proposed for the yeast homolog (Bauerschmitt et al, 2010). The effects of LETM1 deficiency on mtDNA and RNA as well as the 55S ribosome fit with the protein facilitating ribosome assembly, which occurs at the nucleoid (Bogenhagen et al, 2014; Dalla Rosa et al, 2014; He et al, 2012a). On the other hand, yeast mitochondrial ribosomes and nucleoids are interconnected (Kehrein et al, 2015) and copurification analysis is consistent with a similar arrangement in human mitochondria (He et al, 2012b; Rorbach et al, 2008). Thus, LETM1 deficiency could first impair mitochondrial ribosome maintenance and cascade to mtDNA via disturbance of connections between mitochondrial ribosomes and nucleoids.”

2- In Fig. 1, one wonders how much of the LETM1-silencing effect on mitochondrial protein synthesis efficiency relates to lowered levels of mitochondrial ribosomes or to fast degradation of newly synthesized proteins that fail to be inserted into the membrane in the absence of LETM1 (as it occurs in yeast). The protein synthesis experiment should include some shorter pulse times followed by chase if necessary.

The amount of large and small ribosomal subunits decreases with LETM1 knockdown, as do a number of individual ribosomal polypeptides and the ribosomal RNAs (Fig 2). Thus, while we agree with the reviewer that aberrant assembly/insertion of the nascent polypeptides into the membrane might contribute to the OXPHOS defects, the data presented here support a direct effect of LETM1 on mitochondrial ribosomes, and thus mitochondrial protein synthesis.

3- In Fig 3A, ribosomal proteins are shown to co-localize with nucleoid proteins and LETM1. However, this co-localization may not explain the whole story. Questions remaining are:

3.1- is LETM1 interacting with actively transcribing nucleoids?

The reviewers' comments and questions led us to perform a series of immunocytochemistry experiments. We show that LETM1 is not present in discrete foci (either DNA or newly transcribed RNA, or a surrogate (GRSF1)), rather it is more widely distributed, similar to mitochondrial ribosomes or TOM20 (Figure 5).

These results are particularly helpful in 'setting the stage' for the density gradient analyses, as they suggest LETM1's effects on mitochondrial translation relate to interactions with the 55S ribosome as opposed to mtDNA or newly synthesized mtRNA.

3.2- Does LETM1 interacts (or is part of) the mitochondrial nucleoids and the mitochondrial RNA granules too? In fig 4, for example, it would be of interest to stain for RNA.

This question is addressed above, but it also touches on the issue of whether the mtDNA aggregation extends to the mtRNA. Via BrU labeling and GRSF1 immunolabeling after LETM1 KD, we demonstrate that both are markedly affected, in size and distribution, similar to mtDNA (Figure 4, EV1 and Appendix S2). Thus LETM1 deficiency results in mtRNA and RNA granule aggregation.

3.3- In Fig 10, all ribosomes are depicted interacting with the nucleoids; however, the manuscript does not present enough data to support that possibility.

Figure 10 (now Fig 13) has been adapted in light of the reviewer's comments and the new data. In particular, we now emphasize the study of Ott and colleagues showing that yeast mitochondrial nucleoids and ribosomes are linked and cite studies that are concordant with a similar arrangement in mammalian mitochondria. Moreover, the model is helped by the clarification that LETM1 associates primarily with mitochondrial ribosomes rather than mtDNA.

4- What is RETC standing for in Fig 10?

No longer applicable.

5- I would suggest the authors to use the new nomenclature for ribosomal proteins that was recently established (Ban et al)

DAP3 has been changed to MRPS29. If there were any others that did not match the new

nomenclature please advise.

Referee #3 (Comments on Novelty/Model System for Author):

The HeLa model with transient silencing of LETM1 is difficult to interpret. Stable silencing models with clear gene dosage effects would improve the study.

As detailed in the manuscript and described below, the HeLa siRNA analysis produced a series of highly reproducible results with clear gene dosage effects that were fully concordant with the analyses of the fibroblast models of WHS.

Referee #3 (Remarks for Author):

The manuscript by Durigon and colleagues investigates the effects of LETM1 downregulation on mitochondria. LETM1 is one of the chromosome 4 genes that can be deleted in WHS. Therefore, understanding the mitochondrial consequences of LETM1 deficiency may have implications for disease management. LETM1 is thought to play a role in ion regulation in the mitochondrial matrix, but this work potentially highlights different functions. Using both HeLa cells in which the LETM1 gene is partially silenced or primary fibroblasts from WHS patients, the authors show several mitochondrial abnormalities, which include mtDNA-encoded protein expression, mitochondrial nucleoids distribution, mitochondrial fission, and metabolic regulation. The latter is likely the most disease relevant observation, as the mouse model of heterozygote LETM1 KO suffers from glucose utilization problems and severe neurological impairments.

We have rewritten the manuscript in an effort to make it clear to the reader that these different phenotypes are not unconnected but interdependent. This was suggested by 1) the striking mtDNA clustering induced in controls switched to a ketone body diet (Fig 12B-i) and in the same vein; 2) KBs initial accentuate clustering in WHS cells, but this progressively returns to normal (Fig 12B-ii and Appendix Fig S9); 3) in a new experiment we link severe LETM1 deficiency and impaired MPS with metabolic regulation (ketone bodies), in that LETM1 gene silencing in HeLa cells has a less severe effect on MPS in the presence of KB compared to glucose; and finally 4) in another new experiment, we show that the withdrawal of 1 mM pyruvate is sufficient to provoke mtDNA aggregation in control cells.

Hence, the neurological (and developmental problems) of the *Letm1*^{+/-} mouse overlap with WHS, and our data suggest that the glucose/pyruvate utilization problems are inextricably linked to the abnormalities of mtDNA and ribosomes, and therefore just as relevant to the disease. It is of course well established that mtDNA and mitochondrial translation disorders often cause neurological disease.

The results are very interesting and strongly suggest that LETM1 may have roles far more complex than originally described. However, they are also quite preliminary and correlative in nature. At this stage, it is difficult to establish which of the many diverse consequences of LETM1 downregulation described in the different cellular systems are directly and specifically associated with the proposed protein functions and which are secondary. Furthermore, there are several concerns on methodology and data presentation that weaken the conclusions.

These questions and concerns are addressed in the responses to the specific points below.

Specific points

1) Cell culture systems: The relevance of the siRNA experiments in HeLa cells is questionable. The acute siRNA transfection appears to result in inconsistent protein silencing across siRNA and experiments. It is completely unclear how the different results are selected for presentation and result interpretation. At times, the same siRNA appears to silence LETM1 almost completely and at other times less than 50%.

We accept the results could have been presented more clearly, and have revised the text in light of the reviewer's comments. Nevertheless, we maintain that the siRNA experiments in HeLa cells add a considerable amount of value to the study. They establish that LETM1 supports mitochondrial translation and transcript levels, and OXPHOS (in the standard growth conditions where glucose is plentiful), and they help to understand and corroborate the mitochondrial phenotypes seen in WHS

fibroblasts. Moreover, the variation in repression of *LETM1* expression was essential to understand that the levels of residual LETM1 correlates with the OXPHOS deficiency. If all the siRNA experiments had produced severe depletion of LETM1 we would have been concerned about the absence of the phenotype in WHS, whereas the threshold effect in HeLa cells old us not to expect any impairment of MPS in WHS. Prompted by the reviewers comments, we recognized that it would be better on occasion to relate the results to the (residual) level of LETM1 protein rather than detailing every time the particular siRNA used, and this we have now done.

Another example of the value of LETM1 siRNA analyses is DRP1; its inactivation was evident in 6 of 7 contexts (2 (of 3) LETM1 siRNAs and 4 of 4 WHS (*LETM1*^{+/±} cell lines). Thus, while DRP1 repression is a frequent outcome it is not an inevitable consequence of LETM1 deficiency, and moreover the exception (siR1) enabled us to show that dual (*LETM1/DRP1*) knockdown improves MPS and OXPHOS protein levels, cementing the otherwise tentative conclusion that the DRP1 decrease mitigates the effects of LETM1 insufficiency. In the revised manuscript, we make this conclusion the heading of the relevant section.

LETM1 siRNA in HeLa cells supports the conclusion that the mtDNA phenotypes in WHS cells are a consequence of *LETM1* haplo-insufficiency, and in a new experiment we demonstrate that a switch to ketone bodies can mitigate the effects of LETM1 deficiency in respect of mitochondrial protein synthesis (Fig 12D), which speaks to whether the different effects of LETM1 deficiency are linked or reflect distinct functions.

The issue of the time points is also a concern. How certain time points are selected and why is unclear.

The time points 72 and 144 hours relate to one and two rounds of transfection. By and large the phenotypes obtained were very similar, the exception being that the effect of siR2 was more severe after two, as opposed to one, rounds of transfection. Our aim was to show that we had been comprehensive in our analysis, but the reviewer's comments make it clear that this was confusing to the reader. Therefore, we have reduced the amount of data shown in the main figures, and moved some of them to supplementary data for specialist readers.

The primary cells are much more reliable in terms of reproducing the genetics of WHS. However, there is line to line variability, as expected. The authors present results in some of the four lines available in a seemingly random manner. Sometimes Lines 1 and 4 are shown and other times line 3. It is really difficult to determine if the results shown are truly representative of the syndrome as a whole. The study would really benefit of the use of a much better-defined cell system, such as stable cell lines with homo and heterozygote LETM1 KO (e.g., CRISP or TALEN). If the homozygote KO is lethal in cells, as it is in the mouse, this should be at least known. For the fibroblasts, at a minimum, results from all available cell lines should be shown.

We were more comprehensive than we had made clear, and in preparing the revised manuscript we have carried out additional experiments on cell lines S2 and S3, which strengthen the earlier conclusions based on lines S1 and S4. The EM analysis was performed on all the cell lines and the one not previously shown (S2) was atypical in having the most markedly *abnormal* mitochondria of all the WHS lines (revised Fig EV2), demonstrating that we understated rather than exaggerated the extent of the mitochondrial phenotypes. Mitochondrial translation was assayed in all the WHS cell lines and none was different from controls (now included in the revised Appendix Fig S5). Shortly after the EM analysis and protein synthesis assays, the cells of S2 stopped growing, limiting their further study, although in the revision process 1 of 2 remaining vials was used to determine DRP1 activity (phosphorylation of S616) and KB tolerance. DRP1 was repressed similar to S1 and S4, as it was in S3 cells (revised Fig 10 E). S3 cells grew a little better than those of S2, enabling PDH and PDK3 also to be analyzed, both of which were altered in the same way as the earlier analyzed WHS cells (new Fig 11B).

S1 and S4 cells were used for the original cell growth experiments because they had doubling times comparable to controls. In the revised study the analysis has been extended to S2 and S3, and the ability of these cells to tolerate KB has been demonstrated.

Ideally, the phenotypes in both silenced HeLa and fibroblasts should be suppressed by expression of

exogenous LETM1.

Although this can serve as a good confirmation, it is frequently problematic especially when it comes to inner membrane proteins. Our study used multiple cell lines and two approaches (*LETM1* silencing with three different siRNAs (whereas many other studies use only 2) and haplo-insufficiency), which produced a comprehensive set of results that are concordant. Severe depletion of LETM1 is invariably associated with impaired MPS, and reduced transcript levels and low oxidative phosphorylation. Moreover, synthetic reduction of the amount of LETM1 protein by around 50% produces similar mitochondrial alterations to those seen in WHS fibroblasts expressing approximately half the amount of LETM1 protein of controls. Thus, we have demonstrated that these phenotypes are consistent consequences of LETM1 deficiency.

2) The findings described in HeLa cells with severe knockdown of LETM1 appear quite variable depending on which phenotype is analyzed. Overall, there are several diverse consequences, each interesting in itself. However, it is unclear how they relate to each other and how consistent they are. The relationship between mitochondrial morphology and nucleoid distribution, for example, could be one of cause and effect, but which comes first is unknown. The reader is left trying to connect the dots, a daunting task.

The consistent results for severe LETM1 knockdown are impaired OXPHOS, impaired MPS, decreased levels of mitochondrial transcripts and increased mtDNA copy number. Moreover, there was always a change in mtDNA (and RNA) organization, and the differences between siR1 and siR3 can be explained by the difference in DRP1 activity.

The alterations in PDH phosphorylation are presented at the end of the results, but it seems that the authors think that they may be upstream of the changes in DRP1 activation, mitochondrial morphology, and nucleoid distribution. If this is the logical thread, then the metabolic reshaping should be the centerpiece of the work, also in view of the mouse results.

We tried reordering the results along the lines suggested above, but the LETM1 KDs experiments are the basis for assessing whether the WHS cells have mitochondrial abnormalities, so ultimately we reverted to the original basic structure.

3) The study of PDH regulation is underdeveloped and not much new information is revealed, beyond the data that were published by others on the heterozygote mouse. PDH activity should be measured in WHS fibroblasts and in HeLa cells with LETM1 silencing.

We agree our results for PDH are concordant with the changes in glucose metabolism (in fasting conditions) reported for the *Letm1* mouse and contend that this is an important finding. The changes to PDH (phosphorylation of serine 293 of the E1 subunit and increased expression of the corresponding kinase) provide two clear pieces of evidence of its inhibition in WHS, now confirmed in another WHS cell line (Fig 11B). Furthermore, the experiments with ketone bodies corroborate the PDH findings, and they have been reinforced by the new experiment showing that pyruvate withdrawal (in the presence of 0.3 mM BHB and 5 mM glucose) provokes mtDNA aggregation. Therefore, we now present compelling evidence that PDH and pyruvate utilization are altered in WHS. Importantly, the findings strongly suggest the same fundamental alterations occur in mice and humans; and thereby greatly strengthen the earlier supposition that the LETM1 heterozygous mice speak to WHS, which was not established before the current study; and moreover that fibroblasts derived from LETM1-deficient WHS patients are a valid disease model.

It would be interesting to show what would happen if PDH was reactivated by treatment with PDK inhibitors, such as dichloroacetate.

Dichloroacetate (DCA) was at one time tested as a possible therapy for mitochondrial diseases, but exacerbated rather than relieved symptoms; therefore, it may well have unrecognized deleterious effects on mitochondria, beyond activating PDH. In respect of WHS cells, DCA appeared to be slightly less toxic in medium containing 25 mM glucose compared to control cells, but more toxic when glucose was replaced by ketone bodies. This raises some interesting possibilities, but will require considerably more work to understand if DCA's effect on pyruvate utilization is the major reason for the change in growth and survival; therefore we prefer not to include these results in the

manuscript.

[Unpublished data removed at author's request]

4) The expression of the KB utilization pathway is not investigated. Therefore, it is not exactly known how WHS cells adapt to decreased PDH activity.

This is expected to be a major undertaking; it may well involve an uncharacterized phosphorylation or other post-translational modification of a specific factor. A preliminary analysis of expression levels has not revealed a clear candidate.

5) If the hypothesis is correct, lactate production should be increased in HeLa cells silenced and in WHS fibroblasts.

Yes this was strikingly evident in the case of HeLa cells, likely because they are glucose-addicted cancer cells. That said, the *Letm1*^{+/-} mouse shows glycolysis intermediates are up.

[Unpublished data removed at author's request]

6) As fibroblasts with 50% LETM1 do not have mtDNA-encoded protein translation defects or oxidative phosphorylation defects, while HeLa cells with less than 50% do, it is unclear whether LETM1 has two or more independent functions in mitochondria, involving PDH regulation, mitochondrial protein translation, and ion homeostasis. Are these functions related?

7) LETM1 is involved in Ca²⁺ and K⁺ exchange across the inner membrane. It would be important to know if these functions are in any way associated with the phenotypes described here.

In points 6 and 7 the reviewer raises important questions that will need to be answered to complete our understanding of LETM1. Notwithstanding this, we contend that we have produced a comprehensive analysis of LETM1 that establishes its contribution to mitochondrial DNA and RNA organization and protein synthesis, we have shown that a number of these changes can be explained by perceived or actual changes in nutrient availability, including a life and death effect (ketone body tolerance) in the context of LETM1 haploinsufficiency. These findings add substantially to the current knowledge of LETM1, including clarification of its role in WHS, and they provide key mitochondrial functions and behaviors that can be used to resolve the outstanding issues concerning LETM1 modulation of mitochondrial calcium and potassium levels.

Moreover, the pronounced effect of ketone bodies on control cells demonstrates that nutrients can directly affect mtDNA organization (Fig 12A and B); and new experiments link: i) nutrients and MPS, as we show that *LETM1* gene-silenced cells display improved MPS when grown on ketone bodies compared with glucose (Fig 12D); and ii) pyruvate availability in particular impacts mtDNA organization (Fig 12C). These data strongly suggest that multiple functions of LETM1 are related.

8) Several results, especially the cell imaging and the western blots of the fractions, are not quantified. This is important for the dataset for which the authors make statements on significant differences.

Further quantification of the data in figures 2, 4, 7, 10 and 11 has been done.

[Unpublished data removed at author's request]

2nd Editorial Decision

28 May 2018

Thank you for the submission of your revised manuscript to EMBO Molecular Medicine. We have now received the enclosed reports from the referees that were asked to re-assess it. As you will see, while globally supportive, the reviewers took extra time and effort to evaluate the revision and still have a few remaining issues that must be satisfactorily addressed for the paper to move forward.

Further, to gain time should the paper satisfies the 2nd revision criteria, please address editorial points.

Please submit your revised manuscript within two weeks. I look forward to seeing a revised form of your manuscript as soon as possible.

***** Reviewer's comments *****

Referee #1 (Remarks for Author):

Thank you for sending me this revised version of the manuscript by Durigon and colleagues. The authors have certainly taken on board many points raised by the reviewers and I commend them for this and for their additional work. I find the ms a lot clearer and easy to read in the areas that I am more familiar with but I do have some problems with their interpretation of some data related to mitoribosome association. I think this is important, as over speculation may affect others working in the field. The authors show very clearly and nicely that depletion of LETM1 leads to a severe depletion of mitochondrial translation, which is also evidenced by steady state loss of certain OXPHOS markers as might be expected. I am less convinced with what exactly is happening with the mitoribosome. In fig 2A, the ss levels of MRPS17 and to a lesser extent MRPL11 look down in parallel with the loss of LETM1. Fig 2B suggests siR1, more so than siR2 or 3, has a significant effect on levels of 12S and 16S yet on the WB of Fig2A the depletion of LETM1 seems similar for both siR1 and siR3. This is a little odd. In Fig 2C, the total levels of material detected by WB for the NT and siR1 now seems similar when all lanes are included. This is particularly odd when comparing MRPS17 levels in 2A. Finally, when the nontargeting gradients are compared between this figure and Fig 6B (comparing 100mM NaCl) there seems to be major differences in profile particularly for MRPL45. I'm not sure what the error bars represent in Fig 2C ? Mitoribosomal association of LETM1 is essentially being claimed from Fig 6A and B, in particular B. Obviously, there will be some variability when dealing with siRNA depletion expts. However, it is a brave scientist indeed who claims that this data clearly shows the mitoribosome and LETM1 are in association. I am not trying to be too critical of the data, merely the interpretation of the data. These gradients are clearly tricky and therefore making any strong claim, to me, is concerning. I think it important that these concerns be addressed, or if the authors can rebut my comments. To me, the work with ketone bodies and the WHS is of interest and important to the readership of EMM. I am pleased to see the authors focus more on this.

Referee #2 (Comments on Novelty/Model System for Author):

The authors use HeLa cells and fibroblast for mechanistic studies, which is appropriate.

Referee #2 (Remarks for Author):

In the revised version of this manuscript, the authors have made an effort to reduce speculation, and respond to the previous criticism. A role for LETM1 on mitochondrial ribosome stability/assembly is now substantiated by a direct interaction, and the effects of LETM1 deficiency on nutrient usage have been expanded. A large amount of work is presented. The test can still be improved to clarify a few concepts:

1- The concluding section on the interaction of LETM1 with the 55S ribosome reads:

"The effects of LETM1 deficiency on mtDNA and RNA as well as the 55S ribosome fit with the protein facilitating ribosome assembly, which occurs at the nucleoid (Bogenhagen et al, 2014; Dalla Rosa et al, 2014; He et al, 2012a)."

The authors did not take into account in this conclusion the several papers in the literature that have demonstrated that most steps of mitochondrial ribosome assembly occur in the mitochondrial RNA granules. The papers are cited earlier in the text but this should be reiterated in the conclusion.

Also, it is not clear how specific are the effects of LETM1 on ribosome assembly and nucleoid stability. Are the same phenotypes observed in any mutant affecting ribosome proteins or assembly factors? This should be commented in the text.

2- Whether LETM1 is required for mitochondrial ribosome assembly or stability remains unclear. These possibilities could be better explained in the text.

Referee #3 (Remarks for Author):

The authors have nicely addressed many of the previous concerns. They have added numerous clarifications and some new data to improve the manuscript. There are a few relatively minor points (see below), which deserve to be addressed.

Point 6 is the most concerning, as it is not very clear which role LETM1 loss plays in the pathogenesis of WHS. Furthermore, while the functional interaction between LETM1 loss and PDH inactivation is now convincing, the physical relationship with mtDNA nucleoids and the connections with mitochondrial shape remain hypothetical.

1) It is very difficult to appreciate mitochondrial swelling in Fig. 1A. Can this phenomenon be quantified somehow?

2) Which could be the explanation of the apparent disconnect between levels of LETM1 silencing ($R1 > R3$) and mtDNA levels ($R1 < R3$) in Fig. 3?

3) Data in Fig. 6 suggest that a portion of LETM1 may interact physically with the intact ribosome. However, it is likely, also based on immunocytochemistry, that a large portion of LETM1 does not. This possibility needs to be acknowledged.

4) The concept that loss of LETM1 causes mtDNA instability (page 7, first paragraph) needs clarification, as LETM1 silencing leads to increased levels of mtDNA and larger puncta.

5) The mtDNA phenotype in Fig. 7D is unclear. What changes and how by silencing of Drp1 and Drp1+LETM1, respectively?

6) The heterogeneity of LETM1 expression levels in different WHS lines (e.g., S1 vs. S5) and the absence of mitochondrial functional alterations in all affected lines may suggest that the disease occurs independently of LETM1. It is possible that the genetic defect (heterozygote loss of LETM1) is associated with mitochondrial structural abnormalities but not necessarily with WHS.

7) It appears that in silenced HeLa cells mtDNA clustering is associated with impaired MPS possibly by impairing RNA translation. However, whether this is the case also for WHS fibroblasts is still unclear, as apparently these cells have enough LETM1 to prevent MS defects. Therefore, the role of MPS defects in causing disease is unknown.

8) The interaction between nucleoids and PDH is proposed based on the co-fractionation in iodixanol gradient, but this could be due to the complexes being of similar size and density. Furthermore, the degree of co-fractionation in mutant vs. control cells has not been tested here. Therefore, caution should be used in interpreting the result in respect to the cartoon in figure 13.

2nd Revision - authors' response

07 June 2018

Referee #1 (Remarks for Author):

Thank you for sending me this revised version of the manuscript by Durigon and colleagues. The authors have certainly taken on board many points raised by the reviewers and I commend them for this and for their additional work. I find the ms a lot clearer and easy to read in the areas that I am more familiar with but I do have some problems with their interpretation of some data related to mitoribosome association. I think this is important, as over speculation may affect others working in the field. The authors show very clearly and nicely that depletion of LETM1 leads to a severe depletion of mitochondrial translation, which is also evidenced by steady state loss of certain OXPHOS markers as might be expected. I am less convinced with what exactly is happening with the mitoribosome. In fig 2A, the ss levels of MRPS17 and to a lesser extent MRPL11 look down in parallel with the loss of LETM1. Fig 2B suggests siR1, more so than siR2 or 3, has a significant

effect on levels of 12S and 16S yet on the WB of Fig2A the depletion of LETM1 seems similar for both siR1 and siR3. This is a little odd (point 1). In Fig 2C, the total levels of material detected by WB for the NT and siR1 now seems similar when all lanes are included. This is particularly odd when comparing MRPS17 levels in 2A (point 2). Finally, when the nontargeting gradients are compared between this figure and Fig 6B (comparing 100mM NaCl) there seems to be major differences in profile particularly for MRPL45 (point 3). I'm not sure what the error bars represent in Fig 2C? (point 4). Mitochondrial association of LETM1 is essentially being claimed from Fig 6A and B, in particular B. Obviously, there will be some variability when dealing with siRNA depletion expts. However, it is a brave scientist indeed who claims that this data clearly shows the mitochondrion and LETM1 are in association (point 5). I am not trying to be too critical of the data, merely the interpretation of the data. These gradients are clearly tricky and therefore making any strong claim, to me, is concerning. I think it important that these concerns be addressed, or if the authors can rebut my comments. To me, the work with ketone bodies and the WHS is of interest and important to the readership of EMM. I am pleased to see the authors focus more on this.

Point 1: Yes, siR1 and siR3 do both repress LETM1 to similar extents; and both impair MPS and OXPHOS (correlated with decreased steady state levels of structural components) (Figs 1 and 2). Nevertheless, siR1 and siR3 differ in relation to their impact on mtDNA and transcription (reproduced below from figs 2 and 3). Thus, we infer elevated transcription results in rRNA steady-state levels higher than one would expect based on the immunoblots of MRPs. DRP1 activity may well lie at the root of this (Fig. 7), although why DRP1 is co-repressed in every context (siR2, siR3 and 4 x *LETM1*^{+/-} lines) except siR1 is not known.

Point 2: Having been misled for a short period many years ago in respect of another protein, we have learned never to try to estimate protein abundance (MRPs etc.) from immunoblots of material fractionated on sucrose gradients (standard western blots are much more reliable for this purpose).

Point 3: As stated in the figure legends the experiments in fig 2C used whole cell lysates, whereas those in fig 6 were mitochondrial lysates. The latter tend to give to less spread for MRPs. In any case, MRPL45 behaves similarly in all cases: NT siRNA, MRPL45 was concentrated in fractions 8 and 9 (Fig. 2C); in isolated mitochondria of untreated cells (100 mM NaCl) everything sedimented higher on the gradient and thus MRPL45 peaks in fraction 7; and for 50 mM NaCl things are a bit more spread and MRPL45 peaks in fractions 7 and 8 (Fig 6). We now state additionally in the methods that the mitochondrion profiles after gene silencing used whole cell lysates.

Point 4: As stated in the figure legend, n=3 independent experiments, which gave rise to the error bars, that are SEM.

Point 5: We have modified the text to soften the claim that LETM1 is physically associated with the mitochondrial ribosome (suggest replaces indicate); a section heading has been changed to the broader term mitochondrial nucleoprotein complexes; and we have inserted 'a fraction' in the sentence:

... the small amount of intact 55S ribosomes co-migrated with a fraction of the LETM1 protein.

Nevertheless, we think the case is stronger than implied by reviewer 1's comments. A similar

proportion of LETM1 co-fractionated with the mitochondrial nucleoids and ribosomes as TFAM (Figure 6, iodixanol gradients), and from the ICC experiments it is clear that little if any of this LETM1 is bound to nucleoids. In theory, LETM1 could be part of an entirely different complex with the same sedimentation properties as ribosomes and be required for MPS (and normal mtDNA distribution), although there is no such candidate; and we now cite a recent paper that found LETM1 co-purifies with nucleoid proteins and many components of the MPS machinery (Matic et al, 2018), so we are confident that a substantial amount of LETM1 associates with the mitoribosome, directly or indirectly.

Referee #2 (Comments on Novelty/Model System for Author):

The authors use HeLa cells and fibroblast for mechanistic studies, which is appropriate.

Referee #2 (Remarks for Author):

In the revised version of this manuscript, the authors have made an effort to reduce speculation, and respond to the previous criticism. A role for LETM1 on mitochondrial ribosome stability/assembly is now substantiated by a direct interaction, and the effects of LETM1 deficiency on nutrient usage have been expanded. A large amount of work is presented. The text can still be improved to clarify a few concepts:

1- The concluding section on the interaction of LETM1 with the 55S ribosome reads:

"The effects of LETM1 deficiency on mtDNA and RNA as well as the 55S ribosome fit with the protein facilitating ribosome assembly, which occurs at the nucleoid (Bogehagen et al, 2014; Dalla Rosa et al, 2014; He et al, 2012a)."

The authors did not take into account in this conclusion the several papers in the literature that have demonstrated that most steps of mitochondrial ribosome assembly occur in the mitochondrial RNA granules. The papers are cited earlier in the text but this should be reiterated in the conclusion.

We have revised the text such that the conclusion reads:

The effects of LETM1 deficiency on mitochondrial DNA and RNA as well as the 55S ribosome fit with the protein facilitating ribosome assembly, which occurs at the nucleoid (Bogehagen et al, 2014; Dalla Rosa et al, 2014; He et al, 2012a) and closely associated RNA granules (Antonicka et al, 2013; Jourdain et al, 2013; Tu & Barrientos, 2015).

Also, it is not clear how specific are the effects of LETM1 on ribosome assembly and nucleoid stability. Are the same phenotypes observed in any mutant affecting ribosome proteins or assembly factors? This should be commented in the text.

Yes, we fully expect similar effects to be induced by the depletion of other proteins involved in nucleoid organization and ribosome maintenance and we have revised the text as follows:

Whatever the details of the interdependence of mtDNA organization and expression, the findings for LETM1 suggest that insufficiency or defects in many components of mitochondrial nucleoids or the mitochondrial translation machinery could produce similar phenotypes; and indeed MPV17L2 depletion causes marked mtDNA aggregation and severe ribosome maintenance defects (Dalla Rosa et al, 2014); and a mutation in an aminoacyl tRNA synthetase, FARS2, causes mtDNA aggregation (Almalki et al, 2014).

2- Whether LETM1 is required for mitochondrial ribosome assembly or stability remains unclear. These possibilities could be better explained in the text.

55S maintenance covers both aspects, but for improved clarity we have replaced it with the two terms to make it clear to readers that we have not distinguished between them; the revised text reads:

These data suggest that LETM1 is required for 55S ribosome assembly or stability.

Referee #3 (Remarks for Author):

The authors have nicely addressed many of the previous concerns. They have added numerous clarifications and some new data to improve the manuscript. There are a few relatively minor points (see below), which deserve to be addressed.

Point 6 is the most concerning, as it is not very clear which role LETM1 loss plays in the pathogenesis of WHS.

There is a straightforward answer to Reviewer 3's greatest concern (see point 6 below).

Furthermore, while the functional interaction between LETM1 loss and PDH inactivation is now convincing, the physical relationship with mtDNA nucleoids and the connections with mitochondrial shape remain hypothetical.

1) It is very difficult to appreciate mitochondrial swelling in Fig. 1A. Can this phenomenon be quantified somehow?

All reports that have studied *LETM1* knockdown have observed mitochondrial swelling, and so this is merely confirmatory; note also it is easier to appreciate in WHS fibroblasts (Fig. 8B) because fibroblasts have a more distinct and threadlike mitochondrial network. Additionally we have used circularity as a defined measure of swelling and applied this to NT and siR1 treated HeLa cells, the difference is marked and significantly different, as indicated in the revised figure legend and can be seen at a glance in the image below.

2) Which could be the explanation of the apparent disconnect between levels of LETM1 silencing ($R1 > R3$) and mtDNA levels ($R1 < R3$) in Fig. 3?

The difference in DRP1 offers an explanation for the different extent of mtDNA aggregation and equally it could affect the mtDNA levels.

3) Data in Fig. 6 suggest that a portion of LETM1 may interact physically with the intact ribosome. However, it is likely, also based on immunocytochemistry, that a large portion of LETM1 does not. This possibility needs to be acknowledged.

Yes, we agree the possibility needs to be acknowledged, as the ICC results do not distinguish between, none, some or all of the LETM1 being associated with mitochondrial ribosomes. Hence, the importance of the gradient analysis. Reviewer 1 also requested similar changes and the text has

been modified accordingly and a new reference cited of a study that co-purified LETM1 with mitochondrial nucleoprotein complexes (see also the response to reviewer 1).

4) The concept that loss of LETM1 causes mtDNA instability (page 7, first paragraph) needs clarification, as LETM1 silencing leads to increase levels of mtDNA and larger puncta.

We didn't use the term mtDNA instability and so did not introduce the concept that loss of LETM1 causes this problem. Nevertheless, we have revised the section that formed the first paragraph of page 7 in light of another of the reviewer's comments (see point 7 below), and a comment of reviewer 2.

5) The mtDNA phenotype in Fig. 7D is unclear. What changes and how by silencing of Drp1 and Drp1+LETM1, respectively?

To us the combined knockdown of *LETM1* (siR1) and *DRP1* clearly produces more clustering of mtDNA than *LETM1* (siR1) silencing alone and less than *DRP1* silencing alone. We have rearranged the figure so the individual knockdowns are shown first (left to right) and the dual KD is placed on the right hand side (see immediately below).

On the contrary, the data are fully concordant with the mitochondrial functional alterations being dependent on LETM1. The reviewer is correct that fibroblasts of S5 contrasted with the other WHS cell lines in expressing amounts of LETM1 protein equal to controls. But S5 differed from all the other WHS cell lines in one other crucial respect – subject S5 is *LETM1*^{+/+}, like the controls, whereas the other WHS lines are *LETM1*^{+/-}. To increase the prominence of this key piece of information we have modified Fig 8A to indicate the *LETM1* genotypes, and boxed the exceptional case of S5.

A
7) It appears that in silenced HeLa cells mtDNA clustering is associated with impaired MPS possibly by impairing RNA translation. However, whether this is the case also for WHS fibroblasts is still unclear, as apparently these cells have enough LETM1 to prevent MPS defects.

In light of this comment we have reworded the statement at the end of the results section on LETM1 association with mitochondrial nucleoprotein complexes. Given that the loss of one *LETMI* allele in WHS, and the milder *LETMI* silencing, produced mtDNA clustering without evident impairment of MPS, this feature of mitochondria is more vulnerable to *LETMI* insufficiency, and thus cannot be a consequence of impaired MPS per se.

Therefore, the role of MPS defects in causing disease is unknown.

Yes we agree that the mitochondrial dysfunction in WHS does not include impaired MPS, which is why the Discussion focuses on the mitochondrial abnormalities that are common to the knockdowns and the WHS cells (that do not include MPS).

8) The interaction between nucleoids and PDH is proposed based on the cofractionation in iodixanol gradient, but this could be due to the complexes being of similar size and density. Furthermore, the degree of co-fractionation in mutant vs. control cells has not been tested here. Therefore, caution should be used in interpreting the result in respect to the cartoon in figure 13.

We also note in the text that components of PDH co-purify with mitochondrial nucleoids of frogs providing an independent (and conserved) link between the enzyme and mitochondrial nucleoids; and the recent study of MGME1 found the protein co-purifies with PDH and LETM1, at greater significance than several established mtDNA binding proteins, and together with many components of the MPS machinery (Matic et al, 2018), see also the response to reviewer 1. Notwithstanding this we have adopted a more cautious form of words. The revised legend now reads:

Because the bulk of PDH appears to be physically associated with MNPC (Fig 11A).

Corresponding Author Name: Antonella Spinazzola

Journal Submitted to: Embo Mol Med

Manuscript Number: EMM-2017-08550